# The fungal peptide toxin Candidalysin activates the NLRP3 inflammasome and causes cytolysis in mononuclear phagocytes

Lydia Kasper [1], Annika König[1], Paul-Albert Koenig [2], Mark S. Gresnigt [1], Johannes Westman [3], Rebecca A. Drummond [4,5], Michail S. Lionakis[4], Olaf Groß[6], Jürgen Ruland [2,7,8,9], Julian R. Naglik[10] & Bernhard Hube [1,11]

Clearance of invading microbes requires phagocytes of the innate immune system. However, successful pathogens have evolved sophisticated strategies to evade immune killing. The opportunistic human fungal pathogen *Candida albicans* is efficiently phagocytosed by macrophages, but causes inflammasome activation, host cytolysis, and escapes after hypha formation. Previous studies suggest that macrophage lysis by *C. albicans* results from early inflammasome-dependent cell death (pyroptosis), late damage due to glucose depletion and membrane piercing by growing hyphae. Here we show that Candidalysin, a cytolytic peptide toxin encoded by the hypha-associated gene *ECE1*, is both a central trigger for NLRP3 inflammasome-dependent caspase-1 activation via potassium efflux and a key driver of inflammasome-independent cytolysis of macrophages and dendritic cells upon infection with *C. albicans*. This suggests that Candidalysin-induced cell damage is a third mechanism of *C. albicans*-mediated mononuclear phagocyte cell death in addition to damage caused by pyroptosis and the growth of glucose-consuming hyphae.

[1] Department of Microbial Pathogenicity Mechanisms, Leibniz Institute for Natural Product Research and Infection Biology – Hans Knoell Institute, Beutenbergstrasse 11a, Jena 07745, Germany. [2] Institute of Clinical Chemistry and Pathobiochemistry, Klinikum rechts der Isar, School of Medicine, Technical University of Munich, Ismaninger Str. 22, München 81675, Germany. [3] Program in Cell Biology, The Hospital for Sick Children, 555 University Avenue, Toronto, ON M5G 1×8, Canada. [4] National Institute of Allergy and Infectious Diseases, National Institutes of Health, Fungal Pathogenesis Section, Laboratory of Clinical Immunology & Microbiology, 9000 Rockville Pike, Bldg 10 / Rm 11C102, Bethesda, MD 20892, USA. [5] Institute of Immunology and Immunotherapy, Institute of Microbiology and Infection, University of Birmingham, Birmingham B15 2TT, UK. [6] Institute of Neuropathology, Medical Center – University of Freiburg, Faculty of Medicine, University of Freiburg, Breisacher Straße 64, Freiburg 79106, Germany. [7] TranslaTUM, Center for Translational Cancer Research, Technische Universität München, München 81675, Germany. [8] German Cancer Consortium (DKTK), Heidelberg 69120, Germany. [9] German Center for Infection Research (DZIF), Munich 81675, Germany. [10] Mucosal and Salivary Biology Division, King's College London Dental Institute, London SE1 1UL, UK. [11] Friedrich Schiller University, Fürstengraben 1, Jena 07743, Germany. These authors contributed equally: Lydia Kasper, Annika König, Paul-Albert Koenig. Correspondence and requests for materials should be addressed to B.H. (email: bernhard.hube@leibniz-hki.de)

Candida albicans is an opportunistic human pathogenic fungus that causes severe morbidity and mortality in millions of individuals worldwide, with approximately 200,000 deaths attributed to invasive systemic infections each year[1,2]. The ability to undergo a yeast-to-hypha transition is considered one of the main virulence attributes of C. albicans[3] and is accompanied by the expression of infection-associated genes that facilitate adhesion, invasion, nutrient acquisition, host cell damage, biofilm formation, and immune evasion. Consequently, mutant strains locked in the yeast morphology have an attenuated virulence potential to cause systemic infection[3,4]. C. albicans filamentation impacts on fungal recognition by phagocytes (macrophages and dendritic cells (DCs)) of the host innate immune system, activation of pro-inflammatory signalling for host defence, and also on fungal survival and immune escape[5–13].

After recognition of fungal pathogen-associated molecular patterns (PAMPs; e.g., cell wall β-glucan) by phagocyte pattern recognition receptors (PRRs), including Dectin-1[14], C. albicans cells are efficiently phagocytosed by macrophages. Once phagocytosed and contained within a phagosome, C. albicans can still form hyphae, which leads to stretching of phagocyte membranes and host cell killing, thereby facilitating C. albicans' survival and outgrowth[15]. This piercing of host cell membranes by physical forces was thought to be the major pathway of C. albicans immune escape and fungus-induced macrophage damage[9]. However, recent discoveries have led to a paradigm shift in our understanding of C. albicans-phagocyte interactions[16]. Murine-based studies demonstrated that phagocytosed C. albicans induces pyroptosis during early interaction with macrophages, while later events leading to cell damage are mechanistically distinct from pyroptosis, depend on hypha formation[12,17] and are associated with glucose consumption by growing hyphae[18]. Pyroptosis is characterized as an inflammasome-mediated, caspase-1-dependent cell death pathway resulting in IL-1β secretion through pores in the cell membrane, subsequent cell swelling with membrane rupture and, ultimately, cell death[16,19]. Collectively, these data suggest that macrophage killing by C. albicans is a two-stage process, with early pyroptosis-mediated inflammatory damage, followed by physical damage by hyphal piercing[16] and competition for glucose[18].

C. albicans-induced pyroptosis is dependent on NLRP3 (NACHT, LRR, and PYD domains-containing protein 3) inflammasome signalling, a major pro-inflammatory pathway that can integrate multiple cellular stress signals, including those from fungal, bacterial, and viral pathogens or sterile insults[8,20–22]. In general, NLRP3 inflammasome activation requires two sequential events, a priming and an activation step[23–25]. The priming signal (signal 1) is provided by microbial ligands such as fungal β-glucans or bacterial lipopolysaccharide (LPS), leading to the NF-κB-dependent IL1B (pro-IL-1β) and NLRP3 transcription. A subsequent triggering signal (signal 2) activates the inflammasome resulting in the assembly of a multiprotein complex consisting of the sensor protein NLRP3, the adapter protein ASC (apoptosis-associated speck-like protein containing a C-terminal CARD) and the pro-form of the inflammatory protease caspase-1[24–26]. This NLRP3 inflammasome complex serves as a platform for pro-caspase-1 activation and thereby facilitates the processing of its substrates, including pro-IL-1β, for the release of mature bioactive IL-1β[16,21]. Signal 2 can be provided by multiple stimuli, such as extracellular ATP, particulate matter, or viral RNA, but also bacterial pore-forming toxins (PFTs) that activate NLRP3 through still poorly defined mechanisms[25,27,28]. C. albicans hypha formation is known to promote, although not being essential for, inflammasome activation and pyroptosis[7,8,10–13,29]. However, the fungal molecular effectors providing signal 2 are unknown. Furthermore, hypha formation is essential for fungal escape[30] and is required for macrophage lysis by mechanisms distinct from those causing pyroptotic cell death[12].

We recently identified the cytolytic peptide toxin Candidalysin as the missing link between C. albicans hypha formation and host cell damage[31,32]. Candidalysin is encoded by ECE1, one of the core filamentation genes expressed under most hyphae inducing conditions[33], and is therefore exclusively released by C. albicans hyphae, but not yeast cells. ECE1 codes for a polyprotein consisting of eight distinct peptides. After proteolytic processing[34], these peptides, including Candidalysin, are secreted into the extracellular space. Candidalysin is able to directly damage epithelial membranes via membrane intercalation, permeabilisation, and pore formation, causing the release of cytoplasmic constituents[31].

Given the functional similarities to bacterial PFTs[27,28], in this study we dissect the role of Candidalysin in the phagocyte inflammatory and damage response to C. albicans hyphae using a combination of human and murine macrophages and murine DCs. We identify the fungal toxin Candidalysin as a trigger of NLRP3 inflammasome activation and a critical factor required for inflammasome-independent cytolysis.

## Results

**Candidalysin is required for IL-1β release in vivo.** During systemic candidaemia, C. albicans disseminates to vital organs. Organ-specific fungal morphologies and innate immune responses determine if and how C. albicans is cleared in different organs[35]. Given that C. albicans hypha formation[7,8] and bacterial toxins[28] can activate the inflammasome, we hypothesized that the recently discovered hypha-associated cytolytic toxin, Candidalysin[31], can cause IL-1β production, as a key marker of inflammasome activation. Therefore, we investigated the potential of a C. albicans mutant lacking Candidalysin to induce IL-1β production as compared to wild-type (Wt) cells during systemic infection. C. albicans Wt cells infecting kidneys grow predominantly in the hyphal form[35] and high levels of IL-1β were observed (Fig. 1a). In contrast, ece1Δ/Δ mutant cells deficient for Candidalysin[31] showed significantly lower levels of IL-1β responses in the kidney (Fig. 1a). In the spleen, an organ where predominantly yeast cells are observed[35], no significant differences in IL-1β levels were observed between Wt and ece1Δ/Δ infected mice (Fig. 1b). The observation that Candidalysin-deficient ece1Δ/Δ mutants induce significantly lower IL-1β levels

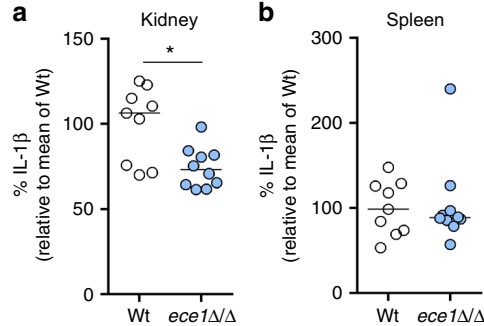

**Fig. 1** Kidney and spleen IL-1β levels during systemic candidemia. **a, b** IL-1β levels measured in **a** kidney and **b** spleen homogenates that were obtained at 1 day post infection from C57/Bl6 mice infected intravenously with C. albicans Wt or the ece1Δ/Δ mutant strain. Values are represented as scatterplot and the median of two independent experiments. The mean of the Wt control group was set at 100% to determine the percentage reduction in IL-1β levels in the mice infected with the ece1Δ/Δ mutant. The means of experimental groups were compared for statistical significance using the Mann–Whitney U test. *p ≤ 0.05

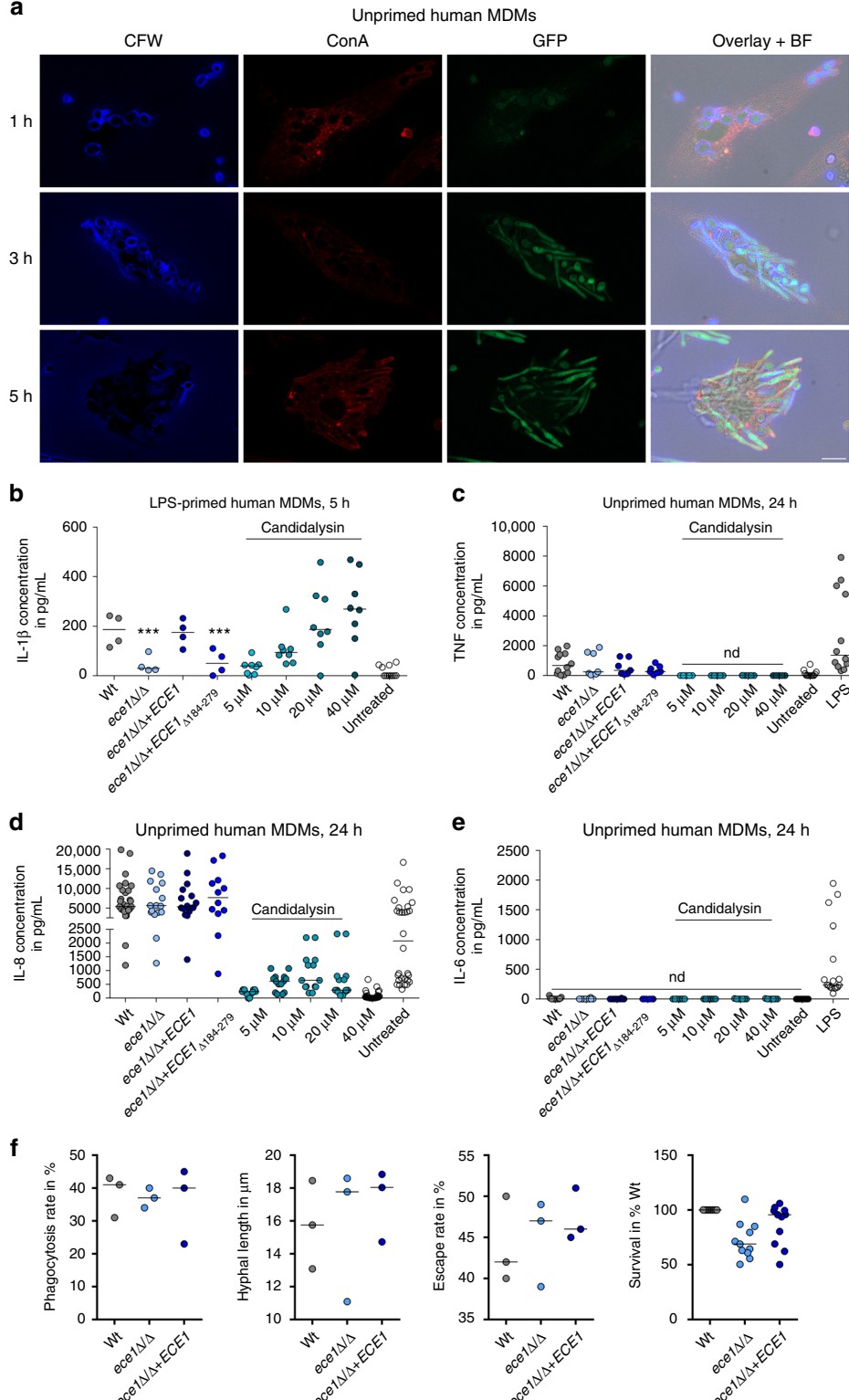

**Fig. 2** (implied) — panels a–f

in an organ where predominantly hyphae are observed highlights an important role for Candidalysin in IL-1β induction.

**Candidalysin induces IL-1β release by human macrophages**. To test whether Candidalysin is a major driver of inflammasome activation in macrophages, we first investigated *ECE1* (coding for Candidalysin) expression using a *C. albicans* reporter strain expressing GFP under the control of the *ECE1* promoter after

phagocytosis by primary human monocyte-derived macrophages (hMDMs) (Fig. 2a). Phagocytosed yeast cells produced hyphae within 3 h and hyphal cells showed a clear GFP fluorescence signal after 3 and 5 h, but not at initial stages (1 h) before hyphal formation was induced. Therefore, *ECE1* is strongly induced in *C. albicans* hyphae after phagocytosis by macrophages.

To study the influence of Candidalysin on inflammasome activation, we measured IL-1β secretion by LPS-primed primary

**Fig. 2** Candidalysin induces IL-1β release by human macrophages. **a** Fluorescence imaging of hMDMs infected with *C. albicans* cells expressing GFP under the control of the *ECE1* promoter. At indicated time points, samples were stained with ConA (non-phagocytosed fungal cells or extracellular hyphae) and Calcfluor White (CFW, phagocytosed and non-phagocytosed fungal cells). Single fluorescence channel images and a composite image of CFW, ConA, GFP, and the bright field (BF) image of one representative experiment out of three are shown. Scale bar 10 μm. **b** IL-1β release measured by ELISA in culture supernatants of LPS-primed hMDMs infected with *C. albicans* Wt, re-integrant (*ece1Δ/Δ + ECE1*) or mutant strains (*ece1Δ/Δ, ece1Δ/Δ + ECE1$_{Δ184-279}$*) (MOI 10) or co-incubated with synthetic Candidalysin for 5 h. **c** TNF, **d** IL-8, and **e** IL-6 release measured by ELISA in culture supernatants of unprimed hMDMs infected with *C. albicans* Wt, re-integrant (*ece1Δ/Δ + ECE1*) or mutant strains (*ece1Δ/Δ, ece1Δ/Δ + ECE1$_{Δ184-279}$*) (MOI 6) or co-incubated with synthetic Candidalysin for 24 h. **f** Phagocytosis rate (1 h *p.i.*), hyphal length of intracellular hyphae (3 h *p.i.*), the rate of hyphae piercing the macrophage membrane (10 h *p.i.*), and and the survival rate of *C. albicans* (3 h *p.i.*, cfus) is shown for human MDMs exposed to *C. albicans* Wt, re-integrant (*ece1Δ/Δ + ECE1*) or mutant strain (*ece1Δ/Δ*) (MOI 1). Values are represented as scatterplot and the median of at least three different donors in at least two independent experiments. For statistical analysis, a one-way ANOVA with Dunnett's multiple comparison test was used. ***$p ≤ 0.001$, nd not detectable. Significance compared to Wt

hMDMs after infection with Wt *C. albicans* and mutants lacking the entire *ECE1* gene (*ece1Δ/Δ*) or only the Candidalysin-encoding sequence (*ece1Δ/Δ + ECE1$_{Δ184-279}$*). Both mutant strains triggered significantly less IL-1β secretion from hMDMs compared to the Wt or to an *ECE1* re-integrant strain (*ece1Δ/Δ + ECE1*) (Fig. 2b). LPS-primed hMDMs stimulated with synthetic Candidalysin also secreted IL-1β in a dose-dependent manner (Fig. 2b). In contrast, secretion of inflammasome-independent cytokines IL-6, IL-8, and TNF from non-primed hMDMs was unaltered when stimulated with the Wt, *ece1Δ/Δ* or *ece1Δ/Δ + ECE1$_{Δ184-279}$* strains. Synthetic Candidalysin induced only low levels of IL-8 and no IL-6, or TNF (Fig. 2c–e). Thus, Candidalysin-deficient *C. albicans* strains exhibit specific defects in IL-1β induction, although they are fully capable of inducing inflammasome-independent pro-inflammatory cytokines in hMDMs.

To understand why the *ece1Δ/Δ* and *ece1Δ/Δ + ECE1$_{Δ184-279}$* mutant strains stimulated much less IL-1β secretion as compared to the Wt, we quantified the influence of *ECE1* deletion on the phagocytosis rate, hyphal length inside macrophages, the rate of hyphal outgrowth from macrophages, and fungal survival after phagocytosis. Deletion of *ECE1* did not influence any of these parameters and no significant differences to the Wt control were observed (Fig. 2f). Therefore, the decreased inflammasome activation in the absence of *ECE1* was not due to reduced uptake of fungal cells or hyphal defects.

An EL4.NOB-1 cell-derived IL-1 bioassay[36] verified that the IL-1β released in the supernatant of human MDMs stimulated by Wt *C. albicans* and the synthetic Candidalysin peptide is indeed bioactive (Fig. 3a). Western blot analyses revealed mature IL-1β in supernatants of LPS-primed phagocytes upon stimulation with *C. albicans* or synthetic Candidalysin (Fig. 3b). Of note, IL-1β secretion was absent in unprimed macrophages stimulated only with Candidalysin (Figs. 3b and 5a, see below). Thus, the priming step (signal 1) is indispensable for Candidalysin-mediated IL-1β production, indicating that Candidalysin selectively provides signal 2 for inflammasome activation. In addition to LPS, a PAMP-derived from gram-negative bacteria, β-glucan-containing molecules, such as Zymosan and Curdlan, were also sufficient as a priming signal for significant IL-1β production (Fig. 3c), which is consistent with the findings of Gross et al.[21] Thus, while dispensable for the priming step of inflammasome induction (signal 1), Candidalysin is a potent trigger of inflammasome activation (signal 2) upon priming with bacterial or fungal PAMPs.

We conclude that Candidalysin is a major activator of the inflammasome and IL-1β secretion in primed hMDMs.

**Candidalysin induces IL-1β release by bone-marrow-derived macrophage (mBMDMs) and bone-marrow-derived dendritic cells (mBMDCs).** To test for the specificity of inflammasome activation in phagocytic cells of different origin, we extended our analysis to primary murine mBMDMs and murine mBMDCs. In contrast to hMDMs, the *ece1Δ/Δ* and *ece1Δ/Δ + ECE1$_{Δ184-279}$* mutants induced Wt-like IL-1β secretion in mBMDMs (Fig. 4a–c), and a moderate, but non-significant reduction in IL-1β induction in mBMDCs (Fig. 4d). However, extracellularly administered synthetic Candidalysin induced a robust, dose-dependent IL-1β response in both mBMDMs and mBMDCs (Fig. 4a–d), whereas secretion of inflammasome-independent TNF in mBMDCs was not affected (Fig. 4e). Therefore, similar to human phagocytes, Candidalysin is able to induce IL-1β secretion from mBMDMs and mBMDCs.

Candidalysin thus acts as a potent inflammasome inducer in both human and murine phagocytes. While Candidalysin alone is sufficient for optimal inflammasome activation in human and murine macrophages and murine DCs, other fungal factors exhibit redundancy in stimulating IL-1β through inflammasome activation in murine phagocytes.

**Candidalysin-activates the NLRP3 inflammasome.** Secretion of IL-1β upon inflammasome activation requires proteolytic processing by caspase-1[26,37]. To investigate whether Candidalysin-triggered processing of pro-IL-1β into mature IL-1β is mediated by caspase-1, we inhibited caspase-1 with the irreversible inhibitors Z-YVAD-FMK or Ac-YVAD-cmk.

Caspase-1 inhibition reduced IL-1β secretion in both *C. albicans*-infected human and murine mononuclear cells after exposure to Candidalysin (Fig. 5a). Yet, both inhibitors did not globally reduce cytokine secretion, because IL-8 or TNF levels were mainly unaltered by Z-YVAD-FMK or Ac-YVAD-cmk treatment (Fig. 5b). Thus, Candidalysin-induced IL-1β secretion is dependent on caspase-1 proteolytic activity. In line with these findings, we observed caspase-1 activation in Candidalysin-treated hMDMs using the fluorescent probe FAM-YVAD-FMK (Fig. 5c). Using a Caspase-GLO assay we detected caspase-1 activity in Wt *C. albicans* stimulated mBMDCs, but significantly reduced caspase-1 activity in mBMDCs exposed to the *ece1Δ/Δ* and *ece1Δ/Δ + ECE1$_{Δ184-279}$* mutants (Fig. 5d). By western blotting, we observed cleaved caspase-1 in culture supernatants of Candidalysin-treated hMDMs as well as mBMDMs and mBMDCs (Fig. 5e). A direct comparison between unprimed and LPS-primed phagocytes showed that the initial priming step is indispensable for Candidalysin-mediated inflammasome activation not only in human macrophages (see above), but also in murine mononuclear phagocytes (Figs. 5a–e and 4c, see above).

Inflammasomes are large protein complexes that include NLR proteins, the adapter protein ASC and pro-caspase-1. Besides NLRP3, which has been demonstrated to be crucial for *C. albicans*-induced inflammasome activation[21,22], several other NLRs, including NLRC4 and NLRP1, trigger the formation of inflammasomes. By using a genetic approach to test whether the

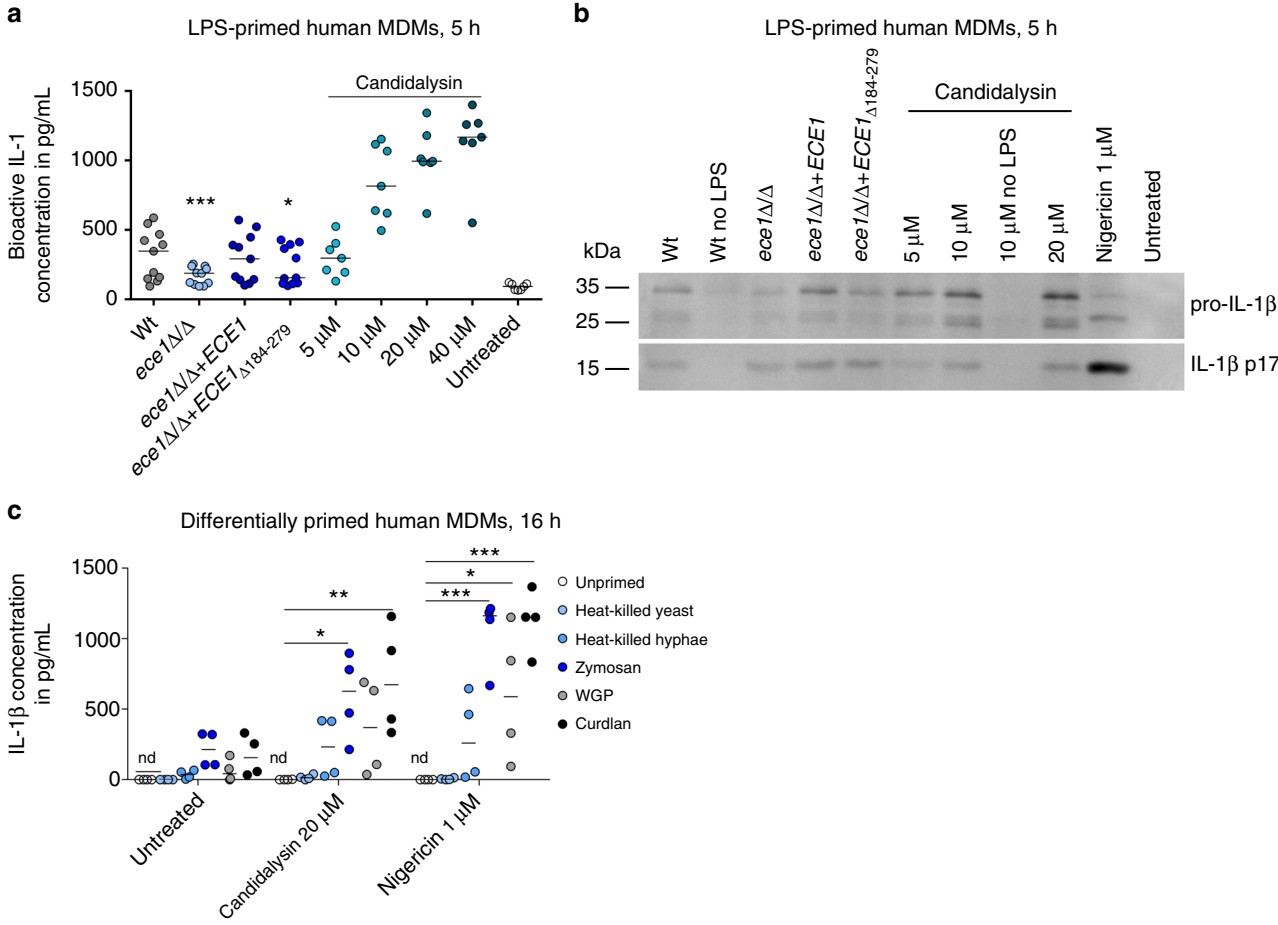

**Fig. 3** Candidalysin-dependent release of bioactive, mature IL-1β by primed hMDMs. **a** Levels of bioactive IL-1 measured in culture supernatants of LPS-primed hMDMs that were infected with *C. albicans* Wt, re-integrant (*ece1Δ/Δ + ECE1*) or mutant strains (*ece1Δ/Δ, ece1Δ/Δ + ECE1$_{Δ184-279}$*) (MOI 10) or co-incubated with synthetic Candidalysin. Bioactive IL-1 was quantified by stimulation of EL4.NOB-1 cells culture supernatants and correlation of the secreted murine IL-2 to a concentration range of recombinant human IL-1β. **b** The presence of processed IL-1β (p17) detected by western blotting in the supernatant of LPS-primed or unprimed (no LPS) hMDMs that were infected with *C. albicans* Wt, re-integrant (*ece1Δ/Δ + ECE1*) or mutant strains (*ece1Δ/Δ, ece1Δ/Δ + ECE1$_{Δ184-279}$*) (MOI 10) or co-incubated with synthetic Candidalysin for 5 h. A representative image of three independent experiments or donors is shown. **c** IL-1β levels were determined by ELISA in culture supernatants of human MDMs that were primed for 16 h with heat-killed *C. albicans* yeasts or hyphae, Zymosan (*Saccharomyces cerevisiae* cell wall), WGP (whole glucan particles; *S. cerevisiae* β-glucan) or Curdlan (β-1,3 glucan) followed by treatment with synthetic Candidalysin or Nigericin for 5 h. Values are represented as scatterplot and the median of at least three different donors in at least two independent experiments. For statistical analysis, a one-way ANOVA with Dunnett's multiple comparison test was used. ***$p ≤ 0.001$, **$p ≤ 0.01$, *$p ≤ 0.05$, nd not detectable. Significance compared to Wt (**a**) or to unprimed cells (**c**)

NLRP3 inflammasome is activated by Candidalysin, we stimulated LPS-primed mBMDCs from *Nlrp3$^{-/-}$*, *Pycard$^{-/-}$* or *Casp1$^{-/-}$* mice[21]. IL-1β secretion was dependent on NLRP3, ASC and caspase-1 respectively (Fig. 5f). Secretion of the inflammasome-independent cytokine TNF was indistinguishable among all tested genotypes (Fig. 5g). These data demonstrate that caspase-1 is fundamentally required for Candidalysin-induced IL-1β secretion via classical NLRP3 inflammasome activation.

**Actin-mediated events and filamentation induce inflammation.** Candidalysin is secreted by *C. albicans* hyphae[31]. Since phagocytes can be exposed to hyphae either pre-phagocytosis or post-phagocytosis, immune cells may be exposed to Candidalysin intracellularly or extracellularly. Therefore, we asked whether internalization of Candidalysin is required for inflammasome activation. hMDMs pre-treated with Cytochalasin D, a well-characterized inhibitor of phagocytosis that impairs actin filament assembly, showed significantly decreased Candidalysin-dependent IL-1β, but not IL-8 secretion (Fig. 6a). In contrast,

IL-1β secretion induced by the potassium ionophore Nigericin was unaffected (Fig. 6a). This suggests that cytoskeletal movement and/or peptide internalization are required for inflammasome activation by Candidalysin and that the mechanism of inflammasome activation by Candidalysin and Nigericin differs.

Candidalysin is necessary for optimal inflammasome activation by *C. albicans* in human macrophages and murine phagocytes (see above). However, deletion of *ECE1* did not completely abrogate IL-1β secretion, indicating that other fungal factors or hypha formation per se (e.g., via physical forces) may be crucial for inflammasome activation[7,8,10-13]. In agreement with this, the *C. albicans* strain *efg1Δ/Δ/cph1Δ/Δ*, which is defective in hyphal formation and the expression of hypha-associated factors[4] induced even lower IL-1β secretion by hMDMs than the *ece1Δ/Δ* mutant (Fig. 6b). However, the *hgc1Δ/Δ* mutant which is defective in hyphal induction, but still can express Candidalysin to some extent[38], induced similar IL-1β levels as the *ece1Δ/Δ* mutant that can form hyphae, but cannot produce Candidalysin (Fig. 6b). Nonetheless, supplementation of synthetic Candidalysin to *efg1Δ/Δ/cph1Δ/Δ*, *hgc1Δ/Δ*, or *ece1Δ/Δ C. albicans* cells

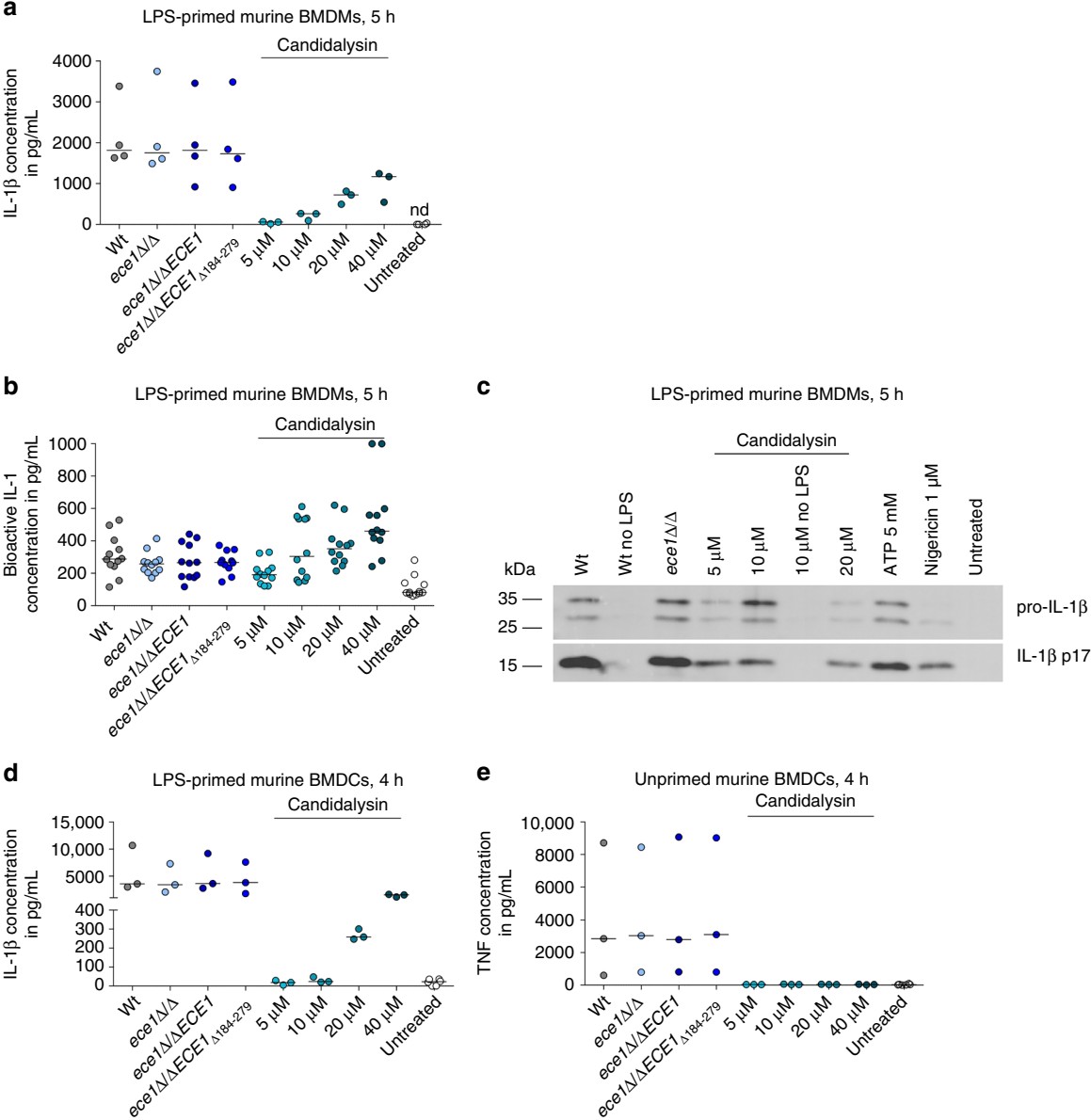

**Fig. 4** Candidalysin induces IL-1β release in murine mononuclear cells. **a** IL-1β release measured by ELISA in culture supernatants LPS-primed mBMDMs infected with *C. albicans* Wt, re-integrant (*ece1Δ/Δ + ECE1*) or mutant strains (*ece1Δ/Δ, ece1Δ/Δ + ECE1_{Δ184–279}*) (MOI 6) or co-incubated with synthetic Candidalysin for 5 h. **b** Levels of bioactive IL-1 measured in culture supernatants of LPS-primed mBMDMs that were infected with *C. albicans* Wt, re-integrant (*ece1Δ/Δ + ECE1*) or mutant strains (*ece1Δ/Δ, ece1Δ/Δ + ECE1_{Δ184–279}*) (MOI 6) or co-incubated with synthetic Candidalysin. Bioactive IL-1 was quantified by stimulation of EL4.NOB-1 cells culture supernatants and correlation of the secreted murine IL-2 to a concentration range of recombinant human IL-1β. **c** The presence of processed IL-1β (p17) detected by western blotting in the supernatant of LPS-primed or unprimed (no LPS) mBMDMs that were infected with *C. albicans* Wt re-integrant (*ece1Δ/Δ + ECE1*) or mutant strains (*ece1Δ/Δ, ece1Δ/Δ + ECE1_{Δ184–279}*) (MOI 10) or co-incubated with synthetic Candidalysin for 5 h. A representative image of three independent experiments or donors is shown. **d** IL-1β and **e** TNF levels measured by ELISA in culture supernatants of LPS-primed or unprimed mBMDCs respectively, that were infected with *C. albicans* Wt, re-integrant (*ece1Δ/Δ + ECE1*) or mutant strains (*ece1Δ/Δ, ece1Δ/Δ + ECE1_{Δ184–279}*) (MOI 5) or co-incubated with synthetic Candidalysin for 5 h (mBMDMs) or 4 h (mBMDCs). Secreted IL-1β (**a**, **d**) and TNF (**e**) were determined by ELISA. Values are represented as scatterplots and the median of at least three different replicates ($n \geq 3$). nd not detectable

restored IL-1β secretion to Wt levels (Fig. 6b). This demonstrates that Candidalysin is necessary for inflammasome activation and can compensate for the lack of other inflammasome-stimulating attributes of *C. albicans*. Interestingly, this compensatory mechanism requires fungal viability, as the rescue effect was not observed with heat-killed *C. albicans* cells as compared to untreated LPS-primed hMDMs.

**Candidalysin activates the inflammasome via K⁺ efflux.** Several mechanisms, such as lysosomal destabilization followed by the release of lysosomal cathepsins, production of reactive oxygen species (ROS), or the permeation of cell membranes leading to ion fluxes are discussed as upstream activators of the NLRP3 inflammasome during fungal infection[39]. To elucidate how Candidalysin triggers inflammasome activation, we first inhibited

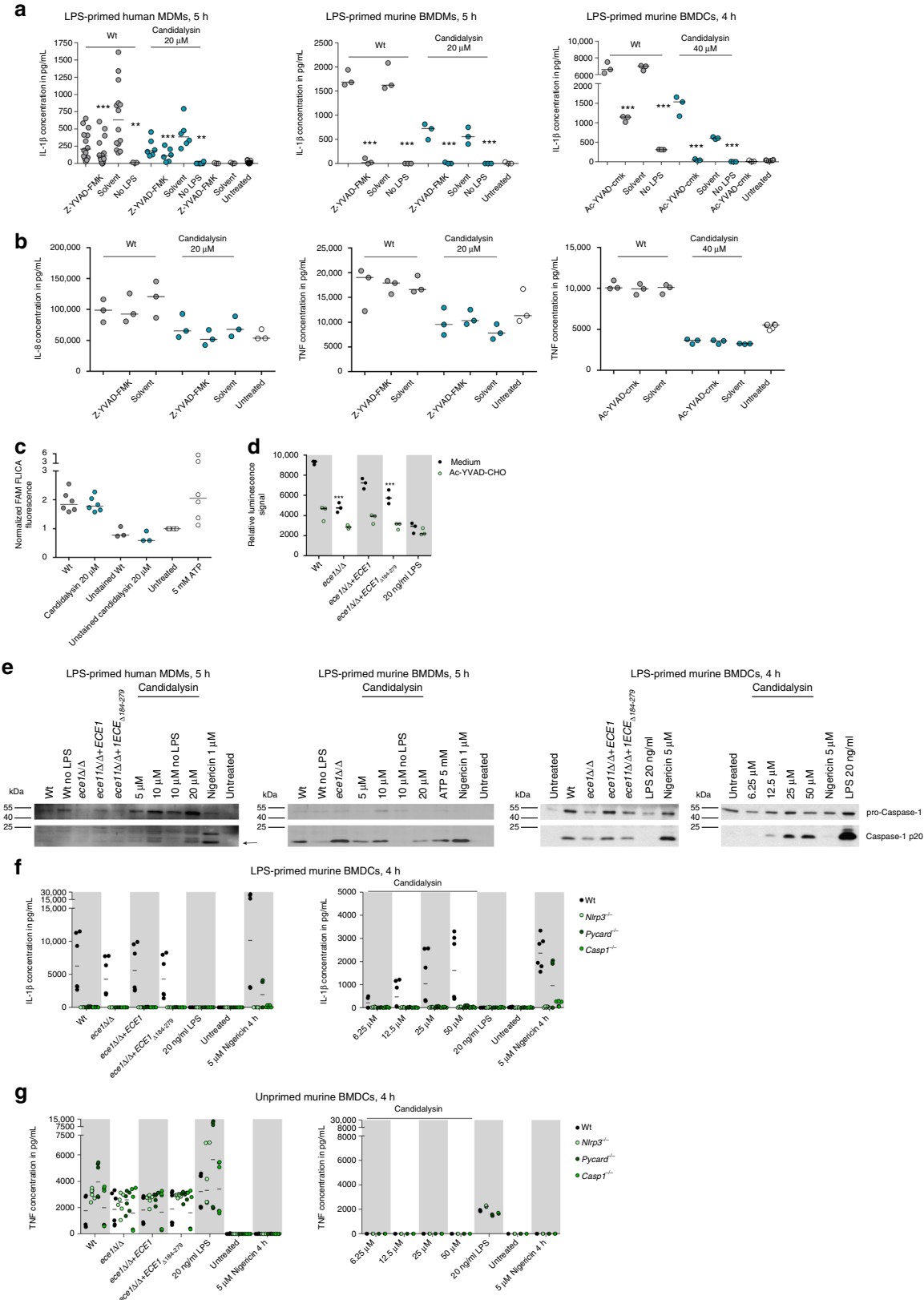

potassium efflux, a common mechanism of inflammasome activation by bacterial toxins and *C. albicans*[21,40]. Inhibition of potassium efflux was achieved by increasing the extracellular

potassium concentration or by blocking ATP-dependent potassium channels with glibenclamide. Similar to the potassium ionophore Nigericin, Candidalysin-dependent IL-1β secretion by

**Fig. 5** Candidalysin activates the NLRP3 inflammasome. **a** IL-1β and **b** IL8 (hMDMs) or TNF (mBMDMs, mBMDCs) release measured by ELISA in culture supernatants of LPS-primed or unprimed (no LPS) hMDMs, mBMDMs, or mBMDCs that were infected with *C. albicans* Wt (MOI 10, 6, or 5 respectively) or co-incubated with synthetic Candidalysin for 5 h (hMDMs, mBMDMs) or 4 h (mBMDCs). The caspase-1-inhibitor Z-YVAD-FMK (88.9 μM, hMDMs and mBMDMs) or Ac-YVAD-cmk (20 μM, mBMDCs) or the inhibitor solute control DMSO was added 1 h prior to infection. **c** Caspase-1 activation measured by fluorescence intensity after staining with FAM-YVAD-FMK FLICA™ in LPS-primed hMDMs that were infected with Wt *C. albicans* (MOI 10), co-incubated with synthetic Candidalysin for 5 h, or treated with ATP for 30 min. **d** Caspase-1 activity measured by luminescence intensity (Caspase1-Glo inflammasome assay) in cell culture supernatants of LPS-primed mBMDCs that were infected for 5 h with *C. albicans* Wt, re-integrant (*ece1Δ/Δ + ECE1*) or mutant strains (*ece1Δ/Δ, ece1Δ/Δ + ECE1_{Δ184−279}*) (MOI 5). **e** Cleavage of caspase-1 into the active p20 form (arrow) assessed by western blotting in LPS-primed or unprimed (no LPS) hMDMs, mBMDMs, or mBMDCs that were infected with *C. albicans* Wt, re-integrant (*ece1Δ/Δ + ECE1*) or mutant strains (*ece1Δ/Δ, ece1Δ/Δ + ECE1_{Δ184−279}*) (MOI 10 mBMDMs, hMDMs or 5 mBMDCs) or co-incubated with synthetic Candidalysin or Nigericin for 5 h (mBMDMs, hMDMs) or 4 h (mBMDCs). Representative images of three independent experiments or donors are shown. **f** IL-1β and **g** TNF levels measured by ELISA in culture supernatants of f LPS-primed or **g** unprimed Wt, *Nlrp3^{−/−}, Pycard^{−/−}* or *Casp1^{−/−}* mBMDCs that were infected with *C. albicans* Wt, re-integrant (*ece1Δ/Δ + ECE1*) or mutant strains (*ece1Δ/Δ, ece1Δ/Δ + ECE1_{Δ184−279}*) (MOI 5) or co-incubated with synthetic Candidalysin or Nigericin for 4 h. Values are presented as scatterplots and the median of at least three different donors or replicates ($n \geq 3$). For KO mBMDCs, all technical replicates are shown of the experiments that were performed in duplicates. For statistical analysis (**a**–**c**), a one-way ANOVA with Dunnett's multiple comparison test was used. ***$p \leq 0.001$, **$p \leq 0.01$, nd not detectable

human MDMs was inhibited by blocking potassium efflux, while IL-8 secretion was not affected (Fig. 6c). In murine BMDMs and BMDCs potassium efflux was similarly important for Candidalysin-dependent IL-1β secretion, but not for TNF secretion (Fig. 6d, e).

Next, we investigated the impact of ROS on Candidalysin-triggered inflammasome activation by inhibiting the NADPH-oxidase-dependent ROS system with (2R,4R)-4-aminopyrrolidine-2,4-dicarboxylate (PDTC). This inhibitor exhibited no effect on Candidalysin-induced IL-1β secretion by hMDMs (Fig. 6f). Consistently, the deletion of *ECE1* or the Candidalysin-encoding sequence alone did not reduce *C. albicans*-induced ROS production in hMDMs, suggesting that Candidalysin does not contribute to fungal ROS induction. Accordingly, ROS levels induced in hMDMs by synthetic Candidalysin are low compared to ROS levels induced by *C. albicans* cells (Fig. 6g, h).

Another mechanism of NLRP3 inflammasome activation involves lysosomal destabilization and lysosomal content release to the cytosol. Proteases such as cathepsins, which require lysosomal acidification to become catalytically active, have been suggested to mediate this effect[39]. Blocking lysosomal acidification with the vacuolar H⁺ ATPase inhibitor Bafilomycin A1 did not reduce Candidalysin-induced IL-1β secretion of hMDMs (Fig. 7a), suggesting that phagosomal destabilization is also not involved in Candidalysin-dependent inflammasome activation. Similarly, co-localization of Wt, *ece1Δ/Δ*, or *ece1Δ/Δ + ECE1_{Δ184−279}* cells with the late endo(lyso)somal marker LAMP1, the late maturation markers Phosphatidylinositol 4-phosphate (PI (4)P) and Rab7[41], as well as with the acidic organelle dye LysoTracker, indicated that phagosome maturation is not affected by Ece1 (Fig. 7a–g). Lastly, administration of synthetic Candidalysin did not lead to a loss of acidification of mature phagosomes loaded with heat-killed *C. albicans* cells as monitored by LysoTracker staining (Fig. 7d). Consistent with the fact that most activators engaging the lysosomal pathway are particles like alum or uric acid crystals, our data indicate that lysosomal mechanisms are not involved in inflammasome activation by Candidalysin. Together, we conclude that induced potassium efflux operates as a main trigger of Candidalysin-induced NLRP3 inflammasome activation comparable to the role of potassium efflux in NLRP3 activation by bacterial PFTs[40].

**Candidalysin is required for damage of hMDMs and mBMDCs**. Previous studies indicate that *C. albicans* causes macrophage damage by two different mechanisms: programmed caspase-1-dependent and inflammation-associated cell death (pyroptosis) within the first hours of infection, followed by physical cell membrane rupture due to sustained hypha formation[9,12,17] and glucose consumption[18] at later time points. As Candidalysin is essential for fungal-induced epithelial cell damage[31], but also activates caspase-1 (see above), we tested whether Candidalysin contributes to *C. albicans*-induced cell damage of mononuclear phagocytes at different time points of infection. By measuring the release of cytoplasmic LDH into the supernatants as a read-out for host cell damage we demonstrate that externally administered synthetic Candidalysin dose-dependently induces cell lysis of human and murine macrophages and murine DCs already at early time-points (Fig. 8a–d). Using human macrophages infected with *C. albicans* for 24 h, we demonstrate that loss of the *ECE1* gene is associated with a loss of the full damage potential of *C. albicans* (Fig. 8a). This coincided with a reduction of metabolic activity of hMDMs (Fig. 8b). LDH levels released from *C. albicans*-infected hMDMs at 5 h were similar to those from an uninfected control. While early *C. albicans*-induced mBMDM damage measured by LDH release did not indicate an *ECE1*-dependency or Candidalysin-dependency (Fig. 8c), damage to mBMDCs induced by *C. albicans* was again partly *ECE1*- and Candidalysin-dependent (Fig. 8d).

To study the damage kinetics of primary hMDMs in more detail, we used propidium iodide (PI) staining to monitor dead immune cells as described in ref. [12]. Similarly, we observed that damage of hMDMs by *C. albicans* Wt (Fig. 8e) occurs in a characteristic biphasic pattern[12]. The first 10–12 h are characterized by a slow increase in host cell damage, whereas in the second phase between 12–24 h damage occurs more rapidly. When *ECE1* or only the Candidalysin-encoding sequence was deleted, the damage potential of *C. albicans* was reduced in both phases in hMDMs (Fig. 8e) highlighting a significant contribution of Candidalysin to *C. albicans*-induced cell damage in human macrophages. Incubation of primary hMDMs with synthetic Candidalysin showed direct, dose-dependent cytotoxicity, as damage (PI-positive cells) occurred rapidly and was saturated within 6 h (Fig. 8f).

In summary, Candidalysin is sufficient to cause rapid damage to both human and murine mononuclear cells and is a major contributor to fungal-mediated damage of hMDMs and mBMDCs.

**Candidalysin-induced cell death is caspase-1-independent**. We observed both Candidalysin-dependent inflammasome activation and early damage of phagocytes. We, therefore, asked whether the inflammatory response and host cell damage in response to Candidalysin are connected and whether cell damage is associated with pyroptosis.

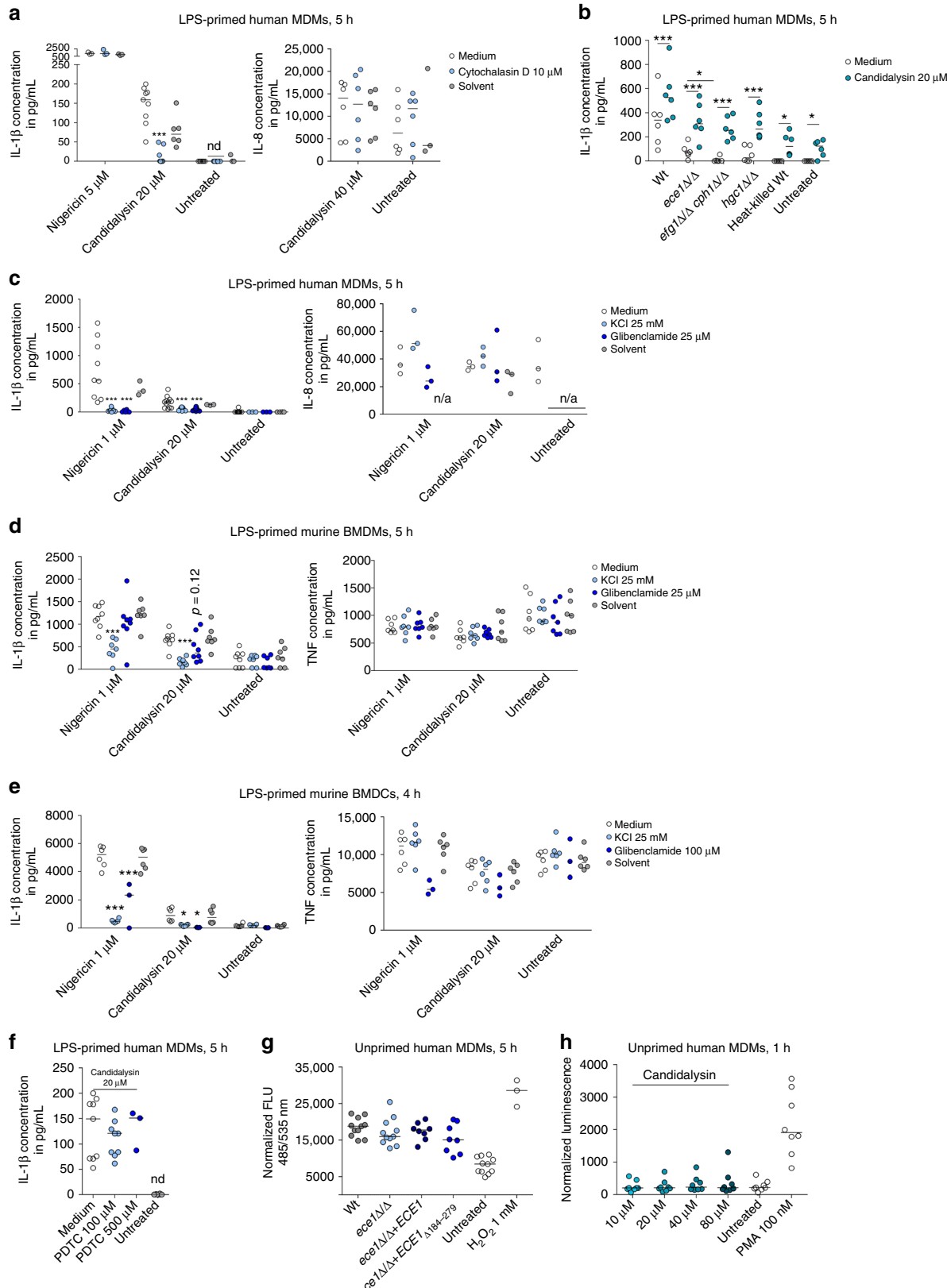

First, to exclude other forms of programmed cell death, we determined whether Candidalysin can induce apoptosis or necroptosis in primary hMDMs. *C. albicans* is able to trigger apoptosis[42] and many bacterial PFTs can induce a programmed form of necrosis, necroptosis[43–45]. Annexin V staining suggested

minimal exposure of cell surface phosphatidylserine in Candidalysin-treated hMDMs and hMDMs infected with Wt or *ece1*Δ/Δ *C. albicans* strains (Fig. 9a). Since Annexin V does not exclusively stain apoptotic but also necroptotic cells, we assayed for the activation of the apoptotic caspases 3 and 7. Both caspases

**Fig. 6** Potassium-dependent and actin-dependent inflammasome activation. **a** IL-1β and IL-8 levels measured by ELISA in culture supernatants of LPS-primed hMDMs treated with synthetic Candidalysin or Nigericin for 5 h. Selected samples were pre-treated with the actin cytoskeleton inhibitor Cytochalasin D or the inhibitor solute control DMSO 1 h prior to administration of Candidalysin. **b** IL-1β release measured by ELISA in culture supernatants of LPS-primed hMDMs that were infected with *C. albicans* Wt, *ece1Δ/Δ*, *efg1Δ/Δ/cph1Δ/Δ*, *hgc1Δ/Δ* mutant strains, or heat-killed Wt (MOI 10) in presence or absence of synthetic Candidalysin for 5 h. **c–f** IL-1β and **c** IL-8 or **d**, **e** TNF levels measured by ELISA in culture supernatants of LPS-primed **c**, **f** hMDMs **d** mBMDMs, and **e** mBMDCs. Phagocytes were treated with synthetic Candidalysin or Nigericin for 5 or 4 h (BMDCs). Selected samples were pre-treated with the following inhibitors 1 h prior to administration of Candidalysin: **c–e** the potassium channel inhibitor glibenclamide or inhibitor solute control DMSO, KCl was added after LPS priming, **f** (2R,4 R)-4-aminopyrrolidine-2,4-dicarboxylate (PDTC). **g** Intracellular ROS production in hMDMs pre-loaded with 20 μM H2DCF-DA for 30 min and infected with *C. albicans* Wt, re-integrant (*ece1Δ/Δ* + *ECE1*) or mutant strains (*ece1Δ/Δ*, *ece1Δ/Δ* + *ECE1*$_{Δ184-279}$) (MOI 10) or treated with $H_2O_2$ (positive control) for 5 h. Fluorescence (Ex 485/Em 535) measured immediately after infection was subtracted from fluorescence (Ex 485/Em 535) measured after 5 h. **h** Total ROS production in hMDMs subjected to synthetic Candidalysin or PMA (positive control) was monitored by Luminol-enhanced chemiluminescence. Relative luminescence units (RLU) were recorded for 60 min and the difference between maximum and minimum luminescence values was calculated. Data are represented as scatterplot and median of at least three different donors ($n \geq 3$) or independent experiments. For statistical analysis, a one-way ANOVA with Dunnett's multiple comparison test was used. For analysis of the different *C. albicans* mutants, a two-way ANOVA with Sidak's multiple comparison test was applied. ***$p \leq 0.001$, *$p \leq 0.05$, n/a not applicable, nd not detectable

were weakly activated upon co-incubation with Candidalysin and no differences were observed when comparing hMDMs stimulated with Wt or *ece1Δ/Δ C. albicans* cells (Fig. 9b). Inhibition of necroptosis with the RIP1-kinase inhibitor Necrostatin-1 also did not diminish macrophage damage (Fig. 9c). Thus, Candidalysin does not appear to trigger apoptosis or necroptosis in human macrophages .

As pyroptosis is characterized by inflammasome activation and subsequent caspase-1-dependent IL-1β secretion[19,46], we measured early macrophage damage in human and murine mononuclear cells after inflammasome priming and the addition of the caspase-1 inhibitor Z-YVAD-FMK. While Caspase-1 inhibition reduced Candidalysin-dependent IL-1β secretion (Fig. 5a, see above), inhibitor treatment had no effect on Candidalysin-induced host cell lysis in hMDMs or mBMDMs, and damage was independent of LPS priming (Fig. 10a, b), though LPS priming was required for cell death of mBMDCs (Fig. 10c). Although previous reports demonstrated that pyroptosis contributes to *C. albicans*-mediated damage of mBMDMs[12,17], LDH levels released by *C. albicans*-infected mBMDMs and mBMDCs were slightly but non-significantly reduced after caspase-1-inhibition (Fig. 10b, c). In line with this, blocking inflammasome activation by inhibiting the host actin cytoskeleton or potassium efflux reduced Candidalysin-induced inflammasome activation (IL-1β release), but not Candidalysin-induced cell damage (Figs. 6a–e and 9d, e, see above).

Damage by Candidalysin is, therefore, mainly independent of inflammasome activation. To exclude that there are differences in the dynamics of *C. albicans* and Candidalysin-induced cell death and to verify our analysis using a different caspase-1 inhibitor, MDMs were LPS-primed and exposed to the caspase-1 inhibitor VX-765. Caspase-1 inhibition did not significantly influence the dynamics of *C. albicans* (Fig. 10d) or Candidalysin (Fig. 10e) induced cell death, although it was effective in reducing inflammasome-dependent IL-1β secretion (Fig. 10f).

Finally, we applied a genetic approach to show that Candidalysin-mediated damage is not pyroptotic. We exposed mBMDCs deficient in the inflammasome components NLRP3, ASC, or caspase-1 to synthetic Candidalysin. Similar to the other immune cell types tested, Candidalysin-induced damage in mBMDCs was independent of LPS-priming, caspase-1, ASC, or NLRP3 (Fig. 10g). As expected, cell lysis induced by live Wt, but also the *ece1Δ/Δ* mutant, *C. albicans* cells was at least partially dependent on the inflammasome, as the overall damage was reduced in *Nlrp3*$^{−/−}$, *Pycard*$^{−/−}$, or *Casp1*$^{−/−}$ as compared to Wt mBMDCs (Fig. 10g). Thus, *C. albicans* lacking Candidalysin can still induce inflammasome-dependent cell death (pyroptosis). Importantly, the reduction in damage caused by the *ece1Δ/Δ* or

*ece1Δ/Δ* + *ECE1*$_{Δ184-279}$ mutant compared to the *C. albicans* Wt was still present in DCs lacking Nlrp3, ASC, or caspase-1.

In summary, these data indicate that *C. albicans*-induced pyroptosis in mononuclear phagocytes is independent of Candidalysin. Moreover, while Candidalysin induces the NLRP3 inflammasome and caspase-1 activation, Candidalysin-induced host cell lysis is independent of the inflammasome and caspase-1.

## Discussion

Phagocytes of the host's innate immune system, such as macrophages and DCs, are pivotally important for efficient clearance of *C. albicans* infections and initiation of inflammatory responses[47]. The cytolytic peptide toxin Candidalysin has recently been identified as a critical virulence factor that intercalates into host membranes and damages epithelial cells during mucosal *C. albicans* infections[31]. Furthermore, Candidalysin drives protective innate type 17 cell responses during oral candidiasis[48], immunopathology during vaginal infections[49], and mediates translocation through intestinal barriers[38].

In this study, using human and mouse mononuclear phagocytes, we show that Candidalysin activates the NLRP3 inflammasome (signal 2 agent), resulting in the secretion of mature IL-1β in a caspase-1-dependent manner. Intriguingly, however, Candidalysin-induced cytolysis is independent of the inflammasome and pyroptosis. Our work identifies Candidalysin as the first fungal toxin with such dual action on phagocytes of the innate immune system.

Inflammasome activation is a two-step process, requiring an initial priming step and a second, inflammasome-activating step[21,23,24]. Our data show that Candidalysin selectively provides a stimulus for the second, inflammasome-activation step, as the toxin alone was not able to induce inflammasome activation without priming by LPS or β-glucan-containing molecules like Zymosan or Curdlan, similar to other NLRP3-inflammasome activators, such as Nigericin or ATP. Multiple stimuli for inflammasome activation, such as mitochondrial damage, ROS production, endo-lysosomal damage, and potassium efflux have been identified[50]. Potassium efflux, in particular, seems to be a central trigger for inflammasome activation for many bacterial PFTs, but also for *C. albicans*[21,40]. We demonstrate that Candidalysin triggers inflammasome activation via potassium efflux in human macrophages, as well as murine BMDMs and BMDCs, suggesting that Candidalysin functions similarly to bacterial PFTs, most likely by inducing membrane permeabilisation and a subsequent drop in cytosolic potassium levels[40,51].

While ROS have previously been implicated in *C. albicans*-dependent inflammasome activation in mBMDCs[21], ROS inhibition with PDTC had no effect on IL-1β secretion in primary

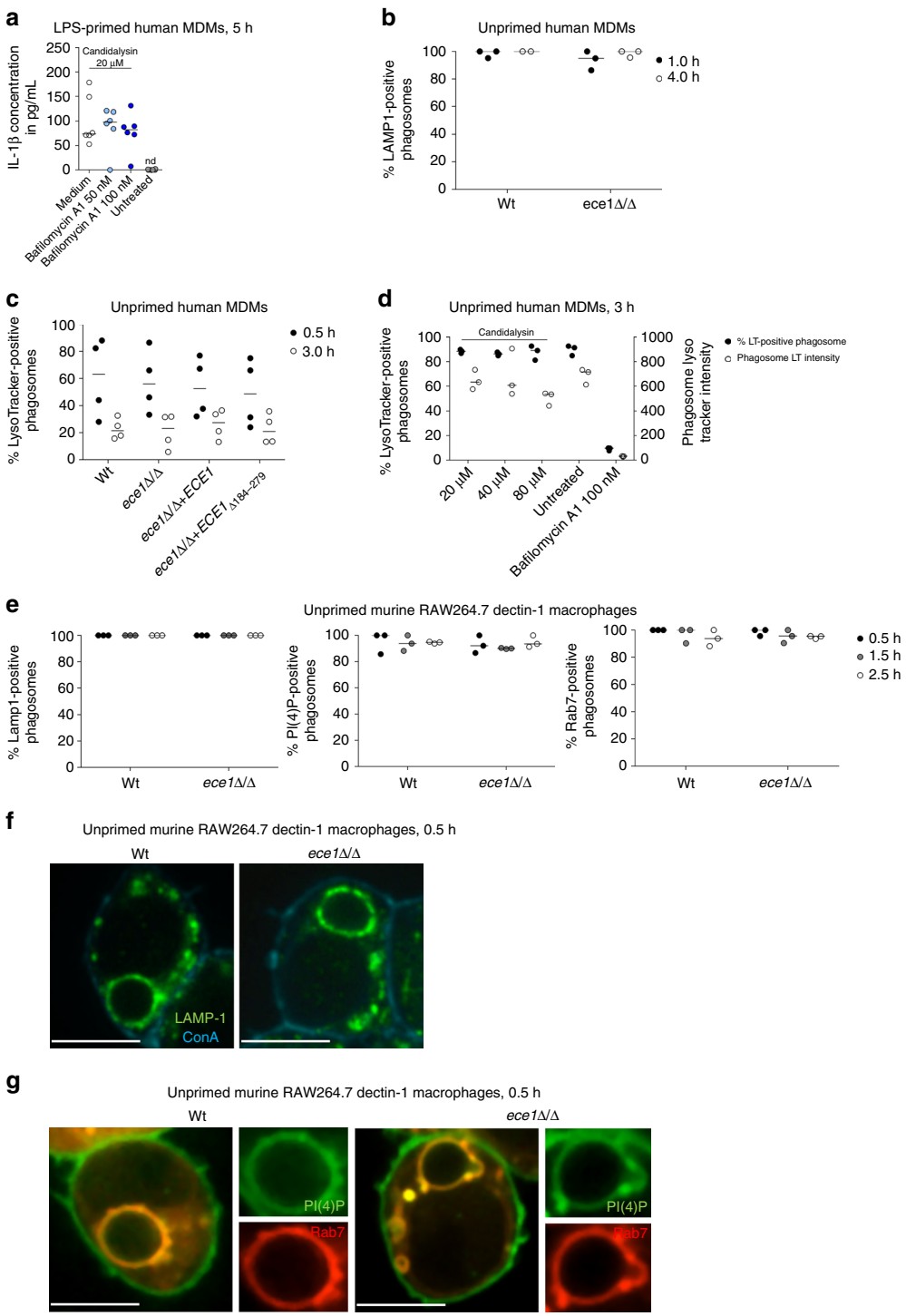

**Fig. 7** Ece1-independent phagosome maturation. **a** IL-1β levels measured by ELISA in culture supernatants of LPS-primed hMDMs. Cells were treated with synthetic Candidalysin for 5 h. Selected samples were pre-treated with the vacuolar H+ ATPase inhibitor Bafilomycin A1 1 h prior to administration of synthetic Candidalysin. **b**, **c** Human MDMs were infected with *C. albicans* Wt, re-integrant (*ece1Δ/Δ + ECE1*) or mutant strains (*ece1Δ/Δ*, *ece1Δ/Δ + ECE1*$_{Δ184-279}$) (MOI 5) and co-localization of *C. albicans*-containing phagosomes with **b** the phagosomal marker LAMP1 or **c** the lysosomal acidification marker LysoTracker was quantified at indicated time points. **d** Human MDMs pre-stained with LysoTracker were infected with heat-killed *C. albicans* Wt (MOI 5) for 3 h in presence or absence of Bafilomycin A1 (phagosomal acidification inhibitor) or synthetic Candidalysin. *C. albicans*-containing phagosomes were quantified for the percentage of LysoTracker-positive phagosomes and LysoTracker intensity. **e** Murine RAW264.7 Dectin-1 macrophages were infected with *C. albicans* Wt or *ece1Δ/Δ* mutant yeasts (MOI 2) and co-localization of *C. albicans*-containing phagosomes with the phago(lyso)somal markers Lamp1, PI(4)P, and Rab7 was quantified at indicated time points. **f**, **g** Murine RAW264.7 Dectin-1 macrophages were infected with *C. albicans* Wt or *ece1Δ/Δ* mutant strain as described in **e**. Representative image of Lamp1 (**f**) or PI(4)P and Rab7 (**g**) acquisition 30 min after phagocytosis. ConA staining of non-phagocytosed *C. albicans*. Scale bar 8 μm. Values are represented as scatterplot with median of three independent donors or experiments (*n* ≥ 3)

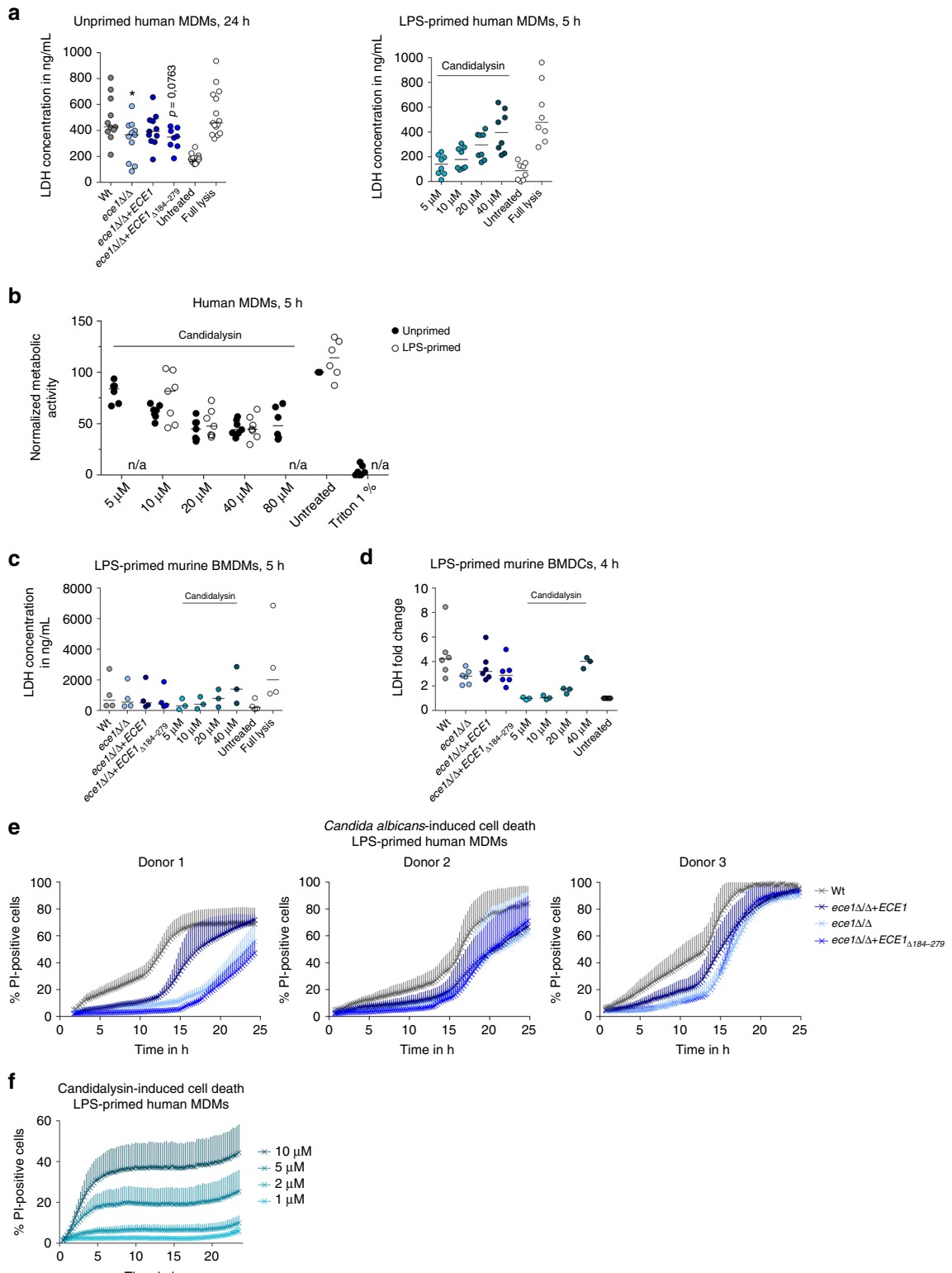

**Fig. 8** Candidalysin-dependent damage of hMDMs. **a**, **c**, **d** Macrophage lysis was quantified by measuring LDH release in **a** unprimed hMDMs or LPS-primed **c** mBMDMs or **d** mBMDCs that were infected with *C. albicans* Wt, re-integrant (*ece1Δ/Δ + ECE1*) or mutant strains (*ece1Δ/Δ*, *ece1Δ/Δ + ECE1_{Δ184-279}*) (MOI 6) for 5 or 24 h and in LPS-primed **a** hMDMs, **c** mBMDMs, or **d** mBMDCs that were incubated with synthetic Candidalysin for 5 h. **b** Metabolic activity of LPS-primed or unprimed hMDMs treated with synthetic Candidalysin for 5 h was measured using XTT. 1% Triton X-100 was added as a positive control. **e**, **f** Macrophage damage over time was assessed by quantifying propidium iodide (PI)-positive cells in LPS-primed hMDMs infected with **e** *C. albicans* Wt, re-integrant (*ece1Δ/Δ + ECE1*) or mutant strains (*ece1Δ/Δ*, *ece1Δ/Δ + ECE1_{Δ184-279}*) (MOI 6) or **f** incubated with synthetic Candidalysin. **a**–**d** Values are represented as scatterplot with median of at least three different donors (*n* ≥ 3). For statistical analysis, a one-way ANOVA with Dunnett's multiple comparison test was used. *$p ≤ 0.05$, significance compared to Wt infection. **e** The results of three different donors are displayed separately due to strong donor variability. Data are shown as mean + SD of two independent positions in two wells. **f** Data are shown as mean + SD of six independent donors

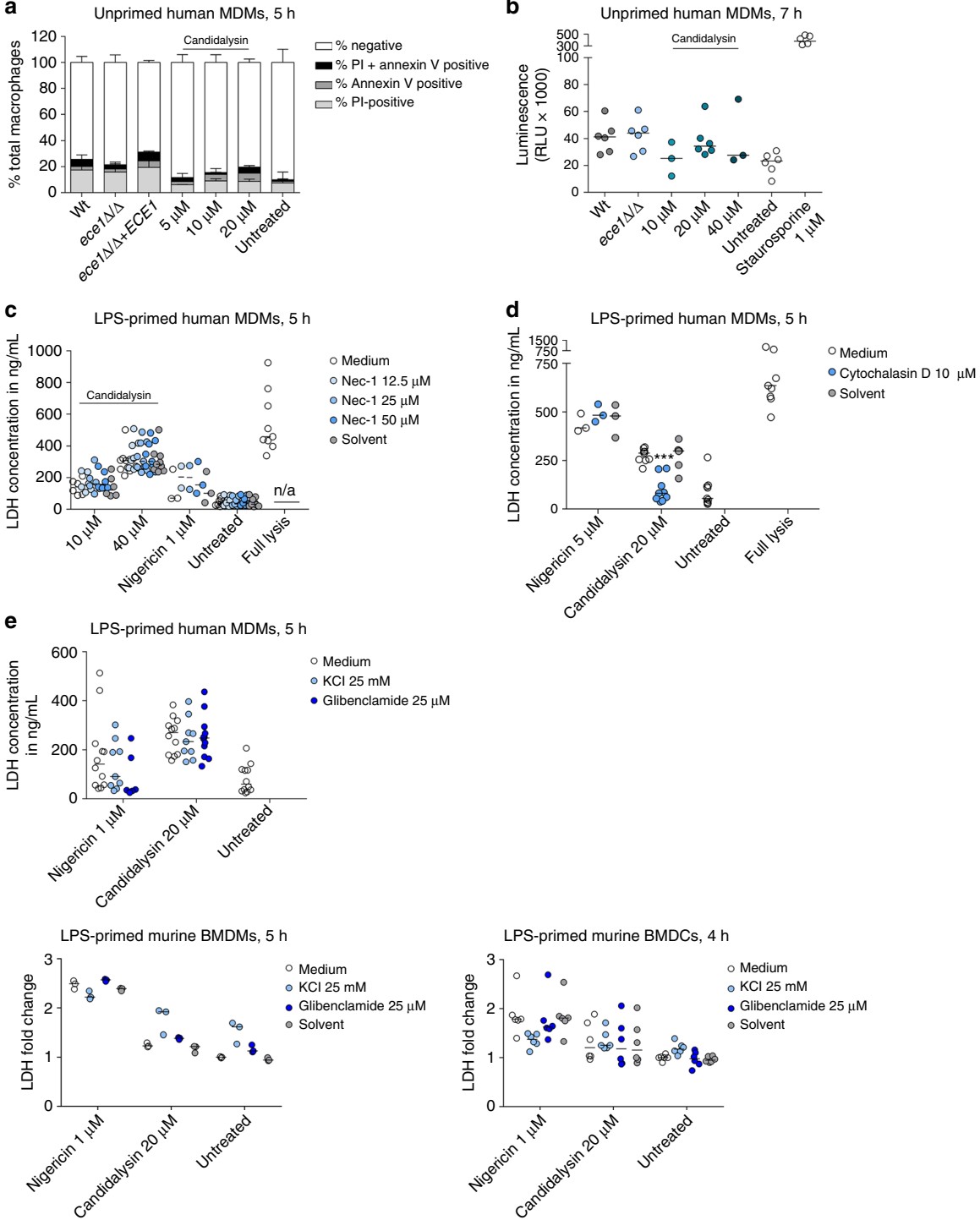

human macrophages. Phagosomal destabilization may also activate the inflammasome, a process thought to involve the release of lysosomal enzymes including cathepsins[52]. However, we found no evidence for phagosomal destabilization and resulting inflammasome activation, which we had hypothesized as a potential result of the intra-phagosomal onset of hypha transformation and lytic activities of Candidalysin produced.

Of note, Candidalysin-dependent inflammasome activation and cellular damage were strongly inhibited by the F-actin polymerisation inhibitor Cytochalasin D. To our knowledge, this is the first description of a pathogen-derived PFT whose

inflammasome activation properties depend on the host cell actin cytoskeleton. In contrast, the ability of bacterial PFTs like Nigericin to activate Nlrp3 is not affected by cytoskeleton inhibitors (this study)[53]. These data suggest that inflammasome activation by Candidalysin may depend on toxin internalization[40,52] or actin-mediated pore-assembly at the cell surface[54].

Our experiments with synthetic Candidalysin peptide isolate the Candidalysin-induced effects from other fungal factors and show a clear role for Candidalysin in inflammasome activation and induction of cell damage in human MDMs, murine BMDMs, and BMDCs. Although analysis of *C. albicans* mutants lacking the

**Fig. 9** Neither apoptosis nor necroptosis is triggered by Candidalysin. **a** Phosphatidylserine exposure and cell viability of hMDMs infected with *C. albicans* Wt, re-integrant (*ece1Δ/Δ + ECE1*) or mutant strain (*ece1Δ/Δ*) (MOI 10) or treated with synthetic Candidalysin for 5 h were quantified by staining with FITC-Annexin V and PI, respectively. The number of single-stained or double-stained macrophages was evaluated by manual counting of at least 200 macrophages. **b** Caspase 3/7 activity was assessed by measuring luminescence of hMDMs 7 h post infection with *C. albicans* Wt or *ece1Δ/Δ* mutant strain (MOI 10) or co-incubation with Candidalysin. Staurosporine served as a positive control. Shown are relative luminescence values (RLU) after background subtraction. **c**, **d** LPS-primed hMDMs were treated with synthetic Candidalysin or Nigericin for 5 h. Selected samples were pre-treated with **c** the necroptosis inhibitor Necrostatin-1 (Nec-1) or **d** the actin cytoskeleton inhibitor Cytochalasin D or inhibitor solute control DMSO 1 h prior to administration of synthetic Candidalysin or Nigericin. Macrophage lysis was quantified by measuring LDH release. **e** LPS-primed hMDMs, mBMDMs or mBMDCs were treated with synthetic Candidalysin or Nigericin for 4–5 h. Selected samples were pre-treated with the potassium channel inhibitor glibenclamide or inhibitor solute control DMSO 1 h prior to administration of synthetic Candidalysin or Nigericin. KCl was added after LPS priming. Macrophage lysis was quantified by measuring LDH release. **a** Data are shown as mean + SD of two different donors. **b–e** Values are represented as scatterplot with median of three independent donors or experiments (*n* ≥ 3). For statistical analysis, a one-way ANOVA with Dunnett's multiple comparison test was used. ***p ≤ 0.001, significance compared to Candidalysin treatment

Ece1- (Candidalysin-) encoding sequence, demonstrated that Candidalysin drives both *C. albicans*-induced inflammasome activation and cellular damage in human macrophages, deletion mutant phenotypes were less prominent or absent in murine BMDMs and BMDCs. This suggests that Candidalysin seems to be more important for human cells as compared to murine cells, but could also be interpreted by the fact that several fungal factors exhibit redundancy in stimulating IL-1β and inducing cell death, particularly in murine phagocytes. Similarly, distinct inflammatory response patterns of murine and human macrophages have been observed when challenged with *Aspergillus fumigatus*[55].

One of such redundant triggers for both, inflammasome activation and damage is likely *C. albicans* filmentation[6,9,56] (this study). Besides, fungal aspartic proteases are known inflammasome inducers[29], and fungal cell wall architecture, ergosterol biosynthesis and phosphatidylinositol-4-kinase signalling play a role in macrophage cytolysis[10–12,17,57].

Importantly, when applying an in vivo systemic candidiasis model, we observed reduced IL-1β levels in mice infected with the *ece1Δ/Δ* mutant as compared to mice infected with wild-type cells. Of note, these differences were only observed in the kidney, an organ where the fungal morphology is dominated by hyphae[35], whereas no differences in the IL-1β release were observed in the spleen, where infecting *C. albicans* cells are predominantly in the yeast morphology. These data highlight that the strictly hyphal associated *ECE1* gene and thus Candidalysin is essential for full IL-1β release during systemic murine infections with *C. albicans*.

The concept that phagocytosed *C. albicans* cells trigger macrophage damage exclusively by mechanical means through sustained filamentation, macrophage membrane stretching and, eventually, host cell lysis, leading to fungal escape[9] has been challenged by a number of recent studies, suggesting a more complex picture of *C. albicans*-macrophage interactions. In murine macrophages, *C. albicans* infection triggers pyroptosis, a regulated inflammatory form of cell death, by activating the NLRP3 inflammasome[12,17]. Pyroptosis is characterized by host cell damage mediated by caspase-1, subsequent pore formation, cell swelling, and eventually membrane rupture[19,46]. Pyroptosis-mediated macrophage damage may thus be an escape route for *C. albicans* within the first six to eight hours of infection before sustained hypha formation results in host cell damage[12,16,17]. However, our data indicate that Candidalysin-induced macrophage lysis is independent of pyroptosis and inflammasome activation, as neither caspase-1 inhibition nor inhibition of potassium efflux nor genetic ablation of caspase-1, Nlrp3, or ASC led to a significant reduction in toxin-induced phagocyte lysis at early time points. In addition, LDH release by phagocytes exposed to the *ece1Δ/Δ* mutant was reduced in inflammasome knockout as compared to wild-type phagocytes. This indicates that pyroptosis still plays a major role in *C. albicans*-induced cell

death by Candidalysin-deficient cells. The bi-phasic cell death dynamics with live *C. albicans* similar to the study of Uwamahoro et al.[12] supports the view that pyroptosis plays a role in the *C. albicans* cell induced cell death. The fact that the predominantly pyroptotic first wave of death[12] is clearly reduced in the *ece1Δ/Δ* mutant, may suggest a minor role for Candidalysin in pyroptosis or that Candidalysin contributes to non-pyroptotic processes in this phase. However, our genetic approach with murine cells lacking key components of the NLRP3 inflammasome clearly demonstrates that Candidalysin induced cell death is predominantly pyroptosis-independent. We can, however, not exclude that Candidalysin, in the setting of live *C. albicans* cells, may facilitate the induction of pyroptosis by other fungal molecules. We also found no evidence for Candidalysin triggering other regulated cell death pathways such as apoptosis or necroptosis. Thus, Candidalysin seems to cause cell death differently from (regulated cell death-inducing) bacterial PFTs such as *Bacillus anthracis* lethal toxin, *Serratia marcescens* hemolysin ShlA, *Clostridium perfringens* β-toxin, or *Staphylococcus aureus* α-hemolysin, while sharing the ability to activate the inflammasome[27,28,43–45,58].

While most known NLRP3 activators including bacterial PFTs kill myeloid cells in an NLRP3 and ASC-dependent manner, there is precedence for NLRP3 activators killing these cells independent of the inflammasome. Three prominent examples are insoluble activators like monosodium urate crystals (MSU) or alum crystals[59], membrane damage by mixed-lineage kinase domain-like protein (MLKL) during necroptosis[60], and cytoplasmic LPS activating Gasdermin D-dependent pyroptosis through caspase-4/11[61]. Similar to Candidalysin, these activators all engage Nlrp3 via K+ efflux, suggesting that membrane perturbations that lead to inflammasome-independent cell death can in parallel activate Nlrp3 through the K+ efflux-mediated mechanism. Furthermore, besides inducing regulated host cell death, Cullen et al.[51] have suggested that signal 2-inducing PFTs, such as streptolysin or listeriolysin, can lead to non-selective permeabilisation of plasma membranes and subsequent necrotic host cell death.

The evidence we have collected so far point to a direct interaction of Candidalysin with host cell membranes as the main cause for toxin-induced necrotic damage.

This study demonstrates that Candidalysin has the ability to damage mononuclear phagocytes and to activate the inflammasome and that these two observations are putatively independent events. Inflammasome activation results in the production of the pro-inflammatory cytokine IL-1β, which, when secreted, induces the recruitment of other immune cells to the site of infection[62,63]. Indeed, the NLRP3 inflammasome has been implicated with an anti-*Candida* response[21,22] and has been shown to induce a protective antifungal Th1/Th17 response[64]. Toxin-dependent inflammasome activation may thus be a disadvantage for the

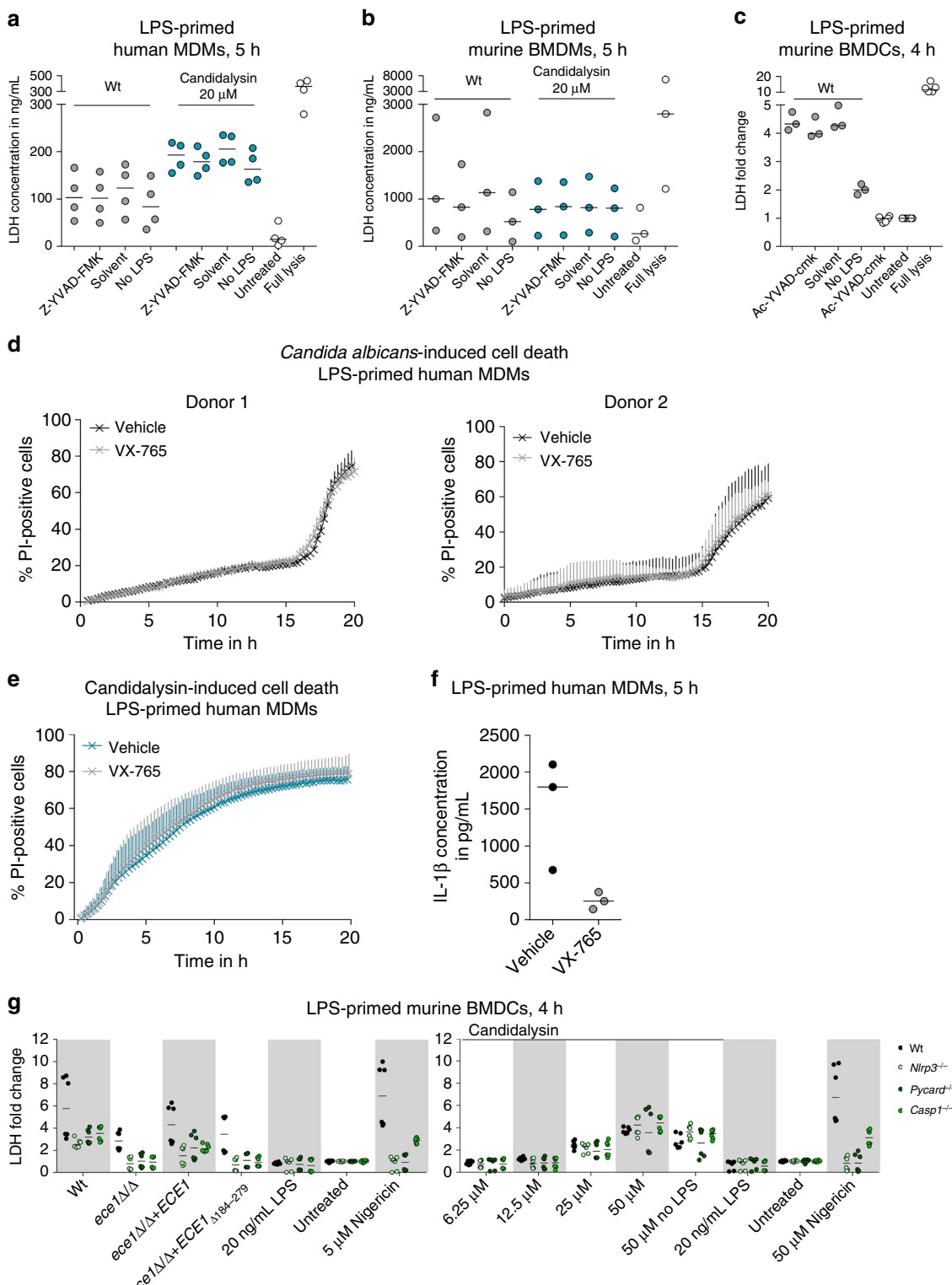

fungus. In contrast, phagocyte damage may be a benefit for the fungus, by supporting immune evasion and escape from macrophage killing by host cell lysis[16].

In light of the literature, we propose the following model for the role of Candidalysin in *C. albicans*-macrophage/DC interaction: The recognition of *C. albicans* PAMPs and/or bacterial ligands of commensal microbes by immune cell PRRs leads to fungal phagocytosis and inflammasome priming. Phagocytosed

fungal cells form hyphae, leading to rapid production of hypha-associated factors such as Candidalysin and other inflammasome-inducing fungal factors. Candidalysin intercalates into host membranes, causing direct plasma membrane permeabilisation leading to ion fluxes that cause a drop in cytosolic potassium. This triggers NLRP3 inflammasome activation and caspase-1-dependent IL-1β processing. Further membrane destabilization ultimately leads to lytic host cell death, thereby contributing to

**Fig. 10** Candidalysin-induced damage is mainly caspase-1-independent. **a–c** Cell damage was quantified by measuring LDH release in LPS-primed or unprimed (no LPS) **a** hMDMs, **b** mBMDMs, or **c** mBMDCs that were infected with *C. albicans* Wt (MOI 6 or 5) for 5 or 4 h respectively or synthetic Candidalysin. The caspase-1-inhibitor Z-YVAD-FMK (88.9 μM, **a**, **b**), Ac-YVAD-cmk (20 μM, **c**) or the inhibitor solute control DMSO was added 1 h prior to infection. **d**, **e** Macrophage damage over time was assessed by quantifying PI-positive cells in LPS-primed hMDMs that were infected with **d** *C. albicans* Wt or **e** incubated with synthetic Candidalysin in the presence or absence of the caspase-1 inhibitor VX-765. **f** Nigericin (5 μm)-induced IL-1β release in LPS-primed hMDMs the presence or absence of the caspase-1 inhibitor VX-765. **g** Cell damage was quantified by measuring LDH release in LPS-primed or unprimed (no LPS) Wt, *Nlrp3*−/−, *Pycard*−/− or *Casp1*−/− mBMDCs that were infected with *C. albicans* Wt, re-integrant (*ece1Δ/Δ + ECE1*) or mutant strains (*ece1Δ/Δ*, *ece1Δ/Δ + ECE1$_{184-279}$*) (MOI 5) or incubated with synthetic Candidalysin for 4 h. **a–c**, **g** Values are represented as scatterplot with median of three independent experiments or donors ($n \geq 3$). **d** The results of two different donors are displayed separately due to strong donor variability. Data are shown as mean + SD of four independent positions in at least 2 wells. **e** Data are shown as mean + SD of six independent donors

the release of mature IL-1β. Concomitantly, caspase-1 activation results in early pyroptotic damage of host cells—a multifactorial process induced in the first hours of infection, which depends on hyphal formation and certain cell wall components[10–12,17,57,65,66], but not Candidalysin. In later phases of infection, mechanical destruction of phagocytes is initiated by hyphae that are formed inside macrophages and pierce the host cell membrane[9]. Hyphal membrane piercing and outgrowth is independent of Candidalysin as mutants lacking Candidalysin have the full potential to escape from host cells.

Our data presented here, collectively with previously published studies on Candidalysin, clearly point towards dual roles of Candidalysin in *C. albicans* pathogenesis, with different outcome depending on the type of infection. First, Candidalysin suits the description of a classical virulence factor[67] that damages host cells. Second, the current study demonstrates that Candidalysin is an immunomodulatory molecule. Such molecules which are sensed by the host immune system to initiate a protective response have been designated as avirulence factors[68,69]. The outcome of the two effects, damage potential vs. protective immune response, dictates the outcome of the infection. During oral infections, epithelial cells recognize Candidalysin via the danger response pathway (via p38 and c-Fos)[31], causing cytokine release and recruitment of phagocytes, in particular neutrophils. This neutrophil recruitment is crucial for pathogenicity, but with oppositional outcome in different tissues (and depending on the immune status of the host). During oral infections, the attraction of neutrophils is protective in immunocompetent mice[48], while neutrophil recruitment during vaginal infections is associated with collateral damage and immunopathology[34]. We believe that similar processes occur in *C. albicans*-infected organs during systemic infections; with macrophages being key players responsible for neutrophil attraction. This concept is, for example, in agreement with the observation that organ-specific fungal morphology and neutrophil attraction correlates with pathogenesis[35].

## Methods

**Ethics statement**. Blood was obtained from healthy human volunteers with written informed consent. The blood donation protocol and use of blood for this study were approved by the Jena institutional ethics committee (Ethik-Kommission des Universitätsklinikums Jena, Permission No 2207–01/08). Animal experiments were performed in compliance with the German animal protection law or approved by the Animal Care and Use Committee of the National Institute of Allergy and Infectious Diseases, USA.

***C. albicans* strains and growth conditions**. *C. albicans* strains included the wild-type (Wt) strain SC5314[70], a derivate of SC5314, a parental strain of the mutant strains used (BWP17-CIp30)[71], an *ECE1* deletion strain (*ece1Δ/Δ*), an *ECE1*-complemented strain (*ece1Δ/Δ + ECE1*), a strain lacking only the Candidalysin-encoding region in Ece1 (*ece1Δ/Δ + ECE1$_{184-279}$*), a *ECE1*-GFP reporter strain (SC5314 + p*ECE1*-GFP; *ECE1* promoter-GFP)[31], and the hypha deficient mutants *efg1Δ/Δ/cph1Δ/Δ*[30] and *hgc1Δ/Δ*[72]. Cells were routinely grown overnight in YPD shaking cultures (1% yeast extract, 2% peptone, 2% glucose) at 30 °C and 180 rpm. Prior to infection experiments, cultures were washed with PBS, counted and adjusted to the desired concentration. *C. albicans* hyphae were prepared by

inoculating PBS-washed yeast cells into RPMI 1640 (Thermo Fisher Scientific) at $6.66 \times 10^6$ cells/mL and incubating for 2 h at 37 °C, 180 rpm. For preparation of heat-killed cells, 500 μL yeast or hyphal cultures were incubated at 70 °C for 10 min.

**Preparation of hMDMs**. Human peripheral blood mononuclear cells (hPBMC) were isolated by Histopaque-1077 (Sigma-Aldrich) density centrifugation from buffy coats donated by healthy volunteers. CD14 positive monocytes were selected by magnetic automated cell sorting (autoMACs; MiltenyiBiotec). To differentiate monocytes into human MDMS (hMDMs), $1.7 \times 10^7$ cells were seeded into 175 cm² cell culture flasks in RPMI 1640 media with 2 mM L-glutamine (Thermo Fisher Scientific) containing 10% heat-inactivated fetal bovine serum (FBS; Bio&SELL) (RPMI + FBS) and 50 ng/mL recombinant human M-CSF (ImmunoTools) and incubated for seven days at 37 °C and 5% $CO_2$. Adherent hMDMs were detached with 50 mM EDTA in PBS, seeded in 6, 24 or 96-well plates to a final concentration of $1 \times 10^6$, $1–2 \times 10^5$ or $4 \times 10^4$ hMDMs/well, respectively in RPMI + FBS and incubated overnight. Macrophage infection experiments were performed in serum-free RPMI medium.

For the differential staining of macrophage phagocytosis and hypha formation after phagocytosis, hMDMs were differentiated by using an adherence method. Briefly, hPBMCs isolated by Histopaque-1077 density centrifugation (see above) were seeded into 100 mm Petri dishes ($4 \times 10^7$ cells/dish) in RPMI 1640 media with 2 mM L-glutamine without FBS and incubated at 37 °C and 5% $CO_2$ for 1–2 h. Following, non-adherent cells were removed by washing twice with PBS. Adherent cells were then differentiated for seven days in RPMI + FBS medium with 50 ng/mL M-CSF as described above.

**Preparation of murine macrophages and DCs**. Murine bone-marrow-derived macrophages (mBMDMs) were generated by culturing bone marrow cells isolated from the femur and tibia of 9 to 19 week old healthy female C57BL/6J mice. For differentiation, $5 \times 10^6$ cells were seeded into a 175 cm² cell culture flask in RPMI + FBS containing 1% Penicillin/Streptomycin (PAA Laboratories) and 40 ng/mL recombinant murine M-CSF (ImmunoTools) and incubated for seven days at 37 °C and 5% $CO_2$. Adherent cells were detached by scraping in RPMI + FBS, seeded in 6 or 24-well plates to a final number of $1.5 \times 10^6$ or $5 \times 10^5$ mBMDMs/well and incubated overnight.

Murine bone-marrow-derived dendritic cells (mBMDCs) were generated by culturing bone marrow cells from 6 to 20 week old C57BL/6J Wt or *Nlrp3*−/−, *Pycard*−/− or *Casp1*−/−[73–75] mice for seven days in mBMDC medium (GlutaMAX-supplemented RPMI + FBS containing 1% Penicillin/Streptomycin (Gibco), 50 μM β-mercaptoethanol (Gibco) and 20 ng/mL recombinant murine GM-CSF (ImmunoTools)). On day 7, the mBMDC culture was harvested. Adherent cells were detached with 5 mM EDTA in PBS, mBMDCs were seeded in 96-well plates in mBMDC medium to a final number of $1 \times 10^5$ mBMDCs/well. Macrophage and DC infection experiments were carried out in serum-free medium.

**Cultivation and transfection of RAW264.7-Dectin-1 cells**. The RAW264.7-Dectin-1-LPETG-3 × HA macrophage cell line (RAW Dectin-1)[76] was grown in RPMI 1640 (Wisent Bioproducts) supplemented with 10% heat-inactivated FBS at 37 °C and 5% $CO_2$ and tested negative for mycoplasma contamination. For transient transfections with plasmids GFP-2xP4M-SidM and iRFP-FRB-Rab7[77], 80% confluent monolayers of RAW264.7 Dectin-1 were collected by scraping and plated onto 1.8 cm glass coverslips at a density of $5 \times 10^4$ cells/coverslip. Macrophages were allowed to recover for 18 h prior to transfection with FuGENE HD (Promega) according to the manufacturer's instructions. Briefly, 1 μg of plasmid DNA and 3 μL of FugeneHD were mixed in 100 μL serum-free RPMI and incubated for 15 min at room temperature. This mix was then distributed equally into four wells of a 12-well plate (Corning Inc.) containing the RAW264.7 Dectin-1 in 1 mL RPMI + FBS. Cells were imaged 18–24 h after transfection.

**Synthetic peptides**. Candidalysin peptide[31] was synthesized commercially (Proteogenix or Caslo). The peptide was dissolved in water and added to phagocytes in concentrations ranging from 1 to 80 μM.

**Infection of hMDMs, mBMDMs, and mBMDCs.** For simultaneous measurement of phagocyte damage and cytokine release 5 h *post infection* (*p.i.*), $2 \times 10^5$ hMDMs or $5 \times 10^5$ mBMDMs/well were seeded into 24-well plates. Murine BMDCs were seeded into 96-well plates to a density of $1 \times 10^5$ mBMDCs/well. For cytokine release or phagocyte damage measurements 24 h *p.i.*, $4 \times 10^4$ hMDMs or $1 \times 10^5$ mBMDCs/well were seeded into 96-well plates. If necessary, phagocytes were primed prior to infection for 2 h (hMDMs, mBMDMs) or 3–4 h (mBMDCs) with 50 ng/mL LPS (Sigma Aldrich). Alternatively, heat-killed yeasts or hyphae (multiplicity of infection (MOI) 10, 100 μg/mL Zymosan (Sigma Aldrich), 100 μg/mL Curdlan (Invivogen) or 100 μg/mL whole glucan particles (WGP dispersible; Invivogen) were used as priming agents. For inhibitor studies, the following compounds were added 1 h prior to infection: the caspase-1-inhibitor Z-YVAD-FMK (88.9 μM; Merck) or Ac-YVAD-cmk (20 μM, Invivogen), the caspase-1-inhibitor VX-765 (50 μg/mL, Invivogen) or vehicle control, the actin cytoskeleton inhibitor Cytochalasin D (10 μM; Sigma Aldrich), the V-ATPase inhibitor Bafilomycin A1 (50–500 nM; Sigma Aldrich), the ROS inhibitor 4-Aminopyrrolidine-2,4-dicarboxylate (PDTC) (100, 500 μM; Enzo Life Sciences), the potassium channel inhibitor glibenclamide (25 μM; Sigma Aldrich) or the RIP1-kinase inhibitor Necrostatin-1 (12.5–50 μM; Biomol). Human MDMs and mBMDMs were infected in 300 μL (24-well plate) or 100 μL (96-well plate) with *C. albicans* at MOI 1 (24 h infection mBMDMs), MOI 6 (24 h infection hMDMs), MOI 6 (5 h infection mBMDMs) or MOI 10 (5 h infection hMDMs) or co-incubated with synthetic Candidalysin. Murine BMDCs were infected in 300 μL in 96-well plates with *C. albicans* at MOI 5 for 4 h or co-incubated with synthetic Candidalysin. Nigericin (1, 5 μM for 4–5 h; Sigma Aldrich), LPS (1 μg/ml; Sigma Aldrich) or ATP (5 mM for 30 min; Invivogen) were used as positive controls. After incubation at 37 °C, 5% $CO_2$ for 4, 5 or 24 h, plates were centrifuged at $250 \times g$ for 10 min and supernatants were harvested.

**IL-1 bioassay.** The murine cell line EL4.NOB-1, which was kindly provided by Prof. L. Joosten (Radboudumc, Nijmegen, The Netherlands), has a high level of surface IL-1 receptor expression, which can recognize both human and murine IL-1. The cell line tested negative for mycoplasma contamination. Constitutively the cells produce practically undetectable IL-2 levels, but in response to bioactive IL-1, the cells produce high concentrations of IL-2. Furthermore, these cells are unresponsive to other cytokines like tumour necrosis factor (TNF), colony stimulating factors (CSFS), IL-3, IL-5, IL-6, and IFN-γ[36]. EL4.NOB-1 cells were seeded in 96 well flat-bottom plates at a final density of $1 \times 10^6$ cells/mL and grown in RPMI 1640 (Thermo Fisher Scientific) supplemented with 10% heat-inactivated FBS at 37 °C and 5% $CO_2$ and were stimulated for 24 h using culture supernatants of hMDMs or mBMDMs stimulated in presence or absence of various concentrations of Candidalysin or *C. albicans* Wt, *ece1*Δ/Δ, *ece1*Δ/Δ + *ECE1* or *ece1*Δ/Δ + *ECE1*$_{\Delta 184-279}$ (as described above). As a control EL4.NOB-1 cells were stimulated with recombinant human IL-1β (R&D systems) in concentrations ranging from 1000 pg/mL to 7.8 pg/mL. After 24 h of incubation at 37 °C, 5% $CO_2$ the culture supernatants were collected and IL-2 production was measured by ELISA (eBioscience), and bioactive IL-1 levels were calculated based on the response to recombinant IL-1β.

**In vivo infections and IL-1β quantification.** Eight week old female C57BL/6 mice (Taconic) were maintained in individually ventilated cages under specific pathogen-free conditions at the 14BS facility at the National Institutes of Health (Bethesda, MD, USA). With 10 mice per group (two independent experiments with five mice each) a power was estimated of 80% ($\beta = 0.80$) with a type I error below 5% ($\alpha = 0.05$) for a variance of 15%. Animals were randomly infected intravenously with $2 \times 10^5$ yeast cells of the indicated fungal strains and humanely euthanized 24 h later for analysis of tissue IL-1β levels. Groups infected with Wt and *ece1*Δ/Δ were unblended for researchers. Kidneys and spleens were aseptically removed and homogenized in PBS supplemented with protease inhibitor cocktail (Roche) and 0.05% Tween 20. Homogenized organs were centrifuged twice to remove debris and resulting supernatants snap-frozen on dry ice and stored at −80 °C prior to analysis. IL-1β concentration in the tissue homogenates was determined by ELISA (R&D Systems), following the manufacturers' instructions.

**LDH-based cell damage assay and cytokine quantification.** Lysis of macrophages and DCs was assayed by measuring the concentration of the cytoplasmic enzyme lactate dehydrogenase (LDH) in cell culture supernatants using the non-radioactive Cytotoxicity Detection or CytoTox-ONE™ Kit (Roche, Promega). Cytokines were quantified in cell culture supernatants by Enzyme-linked Immunosorbent Assay according to the manufacturer's instructions (Ready-SET-Go! ELISA; Thermo Fisher Scientific).

**Quantification of macrophage damage by time-lapse imaging.** For analysis of macrophage cell death kinetics, a method adapted from Uwamahoro et al.[12] was used. Briefly, $6 \times 10^4$ cells/well (hMDMs) were seeded into μ-Slide 8-well chambered coverslips (ibidi) and primed for 2 h with 50 ng/mL LPS prior to infection. Macrophages were then infected with *C. albicans* (MOI 6) or co-incubated with synthetic Candidalysin. Non-phagocytosed yeasts were removed by washing after 1 h. Propidium iodide (PI; 3.33 μg/ml; Sigma Aldrich) was added to stain non-viable

immune cells, chamber slides were transferred to the inverted Zeiss AXIO Observer.Z1 microscope. At least two independent fields/well were imaged every 15 min at 10× magnification for a maximal time span of 24 h using a bright field channel and a DsRed filter. Red channel images were processed using the Fiji software (ImageJ[78]). After conversion to binary images, the number of PI-positive cells was determined using the Particle Analyzer tool. The total number of macrophages was determined manually by counting PI-negative macrophages in an overlay picture of the last time point and adding the Fiji-calculated number of PI-positive macrophages for the same time point.

**XTT assay.** To determine the metabolic activity of Candidalysin-treated macrophages, $4 \times 10^4$ hMDMs/well were co-incubated with synthetic Candidalysin in triplicates in a 96-well plate for 5 h at 37 °C and 5% $CO_2$ in 200 μL RPMI 1640 medium without phenol red (ThermoFisher Scientific). Subsequently, 50 μL of pre-warmed 1 mg/mL XTT and 100 μg/mL coenzyme Q0 (Sigma Aldrich) diluted in RPMI were added and samples were incubated for 2 h at 37 °C. The absorbance at 450 nm was measured with a Tecan Infinite microplate reader, with reference readings at 570 and 690 nm.

**Phagocytosis assay and staining of phagosomes.** $1 \times 10^5$ hMDMs were allowed to adhere to coverslips in a 24-well plate overnight. Acidification of the phagosomes was assessed by adding the acidotropic dye LysoTracker Red DND-99 (Thermo Fisher Scientific; diluted 1:10,000 in RPMI) 1 h prior to infection and during co-incubation with fungal cells. Where indicated, macrophages were pre-treated with 100 nM Bafilomycin A1 (Sigma Aldrich) 1 h before infection. Macrophages were infected with *C. albicans* (MOI 1 to 5) or treated with synthetic Candidalysin. For synchronization of phagocytosis, plates were incubated on ice for 20 min after infection. Unbound yeast cells were removed by washing with RPMI, and phagocytosis was initiated by incubating at 37 °C and 5% $CO_2$. Cells were fixed with 4% paraformaldehyde at the indicated time points. Non-internalized *C. albicans* cells were stained with Alexa Fluor 647-conjugated Concanavalin A (ConA; Thermo Fisher Scientific) for 45 min. For staining of non-internalized and internalized fungal parts, macrophages were permeabilised with 0.5% Triton X-100 in PBS and stained with Calcofluor White (Sigma-Aldrich). For immuno-fluorescence staining of LAMP1, samples were blocked with 5% BSA in PBS after fixation, followed by incubation with a mouse anti-LAMP1 antibody (sc-20011; 1:100; Santa Cruz Biotechnology) for 2 h and with an Alexa Fluor 555-conjugated anti-mouse IgG antibody (A-21424, 1:500; Thermo Fisher Scientific) for 1 h. Coverslips were mounted and fluorescence images were recorded using the Zeiss AXIO Observer.Z1 (Carl Zeiss Microscopy). Phagocytosis and outgrowth rates of intracellular hyphae were calculated by manually counting a minimum of 50 yeast cells/sample. Hyphal length of internalized *C. albicans* cells was measured for 20 cells/sample. The percentage of LAMP1 or LysoTracker-positive phagosomes was evaluated by counting at least 20 or 50 yeast-containing phagosomes/sample, respectively. For evaluation of LysoTracker fluorescence intensities of heat-killed cell-containing phagosomes, the profile option of the Zeiss software ZEN was used. Line-profiles were placed across at least 10 *Candida* cells/sample and intensity peaks of DsRed channel images were recorded.

For analysing phagosomal maturation in murine macrophages, RAW264.7 Dectin-1 macrophages were infected with *C. albicans* Wt or *ece1*Δ/Δ mutant (MOI 2). Yeast cells were centrifuged onto macrophages at $300 \times g$ for 1 min and phagocytosis was allowed for 20 min at 37 °C and 5% $CO_2$, before non-adherent cells were removed. Remaining non-phagocytosed yeasts were outside-labelled with Alexa Fluor 647-conjugated ConA for 10 min. All outside-labelled yeast cells were excluded from the experiment. At given time points, macrophages were fixed in ice-cold 100% methanol for 5 min at −20 °C, followed by extensive washing in PBS. Phagolysosomes were detected using rat anti-Lamp1 hybridoma (1D4B, 1:20, Developmental Studies Hybridoma Bank), and visualized using a donkey anti-rat Alexa Fluor 488-coupled secondary antibody (712-545-150, 1:1000, Jackson ImmunoResearch). For quantification of Phosphatidylinositol 4-phosphate (PI(4)P) and Rab7 acquisition, RAW264.7 Dectin-1 macrophages were transiently co-transfected with GFP-2xP4M (PI(4)P binding domain) and RFP-Rab7. At given time points, micrographs were acquired using a spinning-disk confocal microscope (Quorum Technologies), and at least 16 Lamp1-, PI(4)P- and Rab7-positive phagosomes were quantified using Volocity 6.3 (Perkin Elmer Inc.), counting at least 16 phagosomes/sample.

**Survival assay.** $4 \times 10^4$ hMDMs were seeded in 96-well plates in RPMI + FBS containing 100 U/mL IFN-γ (Immunotools), infected with *C. albicans* (MOI 1) and incubated at 37 °C and 5% $CO_2$. The assay was performed in triplicates. The survival of yeast cells internalized by macrophages was assessed after 3 h by removing non-hMDM-associated fungal cells by washing with RPMI, subsequent lysis of hMDMs with 20 μL 0.5% Triton X-100 per well and plating lysates on YPD plates to determine fungal burdens (colony forming units (cfus)). Lysate cfus were normalized to cfu numbers of the respective inoculum.

**Caspase-1 activation assay.** Caspase-1 activation in hMDMs was assayed using the FAM FLICA™ caspase-1 Kit (Bio-Rad). Briefly, $1 \times 10^5$ hMDMs/well were seeded into 24-well plates containing glass coverslips and incubated overnight at

37 °C and 5% $CO_2$. Prior to infection, macrophages were primed for 2 h with 50 ng/mL LPS. Cells were then infected with *C. albicans* (MOI 10) or subjected to synthetic Candidalysin. Non-phagocytosed fungal cells were removed after 1 h by washing. After 4 h, FAM-YVAD-FMK FLICA™ reagent was added to a final concentration of 1× and macrophages were incubated for an additional 1 h. Subsequently, nuclei were stained with Hoechst, samples were fixed according to the manufacturer's instructions and fluorescence images were recorded by a Leica DM5500B microscope, using appropriate filters for the detection of FAM FLICA (green channel) and Hoechst (DAPI channel) signals. Fluorescence intensity of FAM FLICA in macrophages was quantified using the quantification tool for region of interest (ROI) of the Leica LAS AF microscope software. ROIs were placed around macrophages and mean grey values were recorded. Background values (ROI placed in region without macrophages) were subtracted.

Caspase-1 activity in mBMDCs was determined using the CaspaseGlo 1 Inflammasome assay (Promega). $1 \times 10^5$ mBMDCs/well were seeded into 96-well plates and primed with LPS for 3 h or left untreated for 3 h before infection with *C. albicans* (MOI 5) for 5 h. For the CaspaseGlo 1 inflammasome assay, supernatant were transferred to white 96-well plates and mixed with the supplemented substrate mix in the presence or absence of the Caspase-1 inhibitor Ac-YVAD-CHO, according to the manufacturer's manual. The plotted values were detected 60 min after mixing samples and the supplemented substrate mixes. Blank values of medium without cells were subtracted from sample values.

**Annexin V-based cell death assay**. Phosphatidylserine exposure on hMDMs was quantified using FITC-Annexin V (Biolegend), according to the manufacturer's instructions. Briefly, $1 \times 10^5$ hMDMs/well were seeded into 24-well plates containing glass coverslips and incubated overnight at 37 °C and 5% $CO_2$. Cells were infected with *C. albicans* (MOI 10) or treated with synthetic Candidalysin and incubated for 3 or 7 h; Staurosporine (1 μM; Sigma Aldrich) was used as a positive control. Cells were incubated with 5 μL FITC-Annexin V and 5 μg/mL PI (Sigma Aldrich) in 200 μL annexin binding buffer for 15 min, mounted with DAPI and imaged immediately using the Zeiss AXIO Observer.Z1 (Carl Zeiss Microscopy). Images were evaluated manually for Annexin V- and PI-positive cells by counting at least 200 macrophages.

**Caspase-3/7 activation assay**. Caspase-3/7 activity in hMDMs was determined using the Caspase-Glo 3/7 Assay (Promega). $4 \times 10^4$ hMDMs/well were seeded into white clear-bottom 96-well plates and incubated overnight at 37 °C and 5% $CO_2$. Macrophages were infected with *C. albicans* (MOI 10) or treated with synthetic Candidalysin and incubated for 7 h. Staurosporine (1 μM; Sigma Aldrich) was used as a positive control. Caspase-Glo substrate was added and cells were incubated at room temperature for 60 min. Luminescence was recorded using a Tecan Infinite microplate reader.

**Luminol-based ROS detection**. Total ROS production by hMDMs was quantified by chemiluminescence. Briefly, $4 \times 10^4$ hMDMs/well were seeded into white clear-bottom 96-well plates and incubated overnight at 37 °C and 5% $CO_2$. All cells and reagents were prepared in RPMI 1640 without phenol red. Cells were subjected to synthetic Candidalysin or 100 nM PMA as a positive control. All samples were prepared in triplicates. Fifty microliters of a mixture containing 200 mM luminol and 16 U horseradish peroxidase were added immediately prior to quantification. Luminescence was measured every 3 min over a 60 min incubation period at 37 °C using a Tecan Infinite microplate reader. For each sample, minimum and maximum luminescence values were determined and the difference was calculated (MAX–MIN).

**Intracellular ROS measurement ($H_2$DCF-DA)**. $4 \times 10^4$ hMDMs/well were seeded into black clear-bottom 96-well plates and incubated overnight at 37 °C and 5% $CO_2$. Immediately before infection, cells were loaded with 20 μM $H_2$DCF-DA (Sigma Aldrich) in pre-warmed PBS for 30 min at 37 °C, 5% $CO_2$ and washed with pre-warmed PBS. Macrophages were then infected with *C. albicans* (MOI 10) or treated with $H_2O_2$ (1 mM) or PMA (1 μM). Fluorescence (Ex 485/Em 535) was recorded immediately after infection and after 5 h incubation at 37 °C and 5% $CO_2$ using a Tecan Infinite microplate reader and fluorescence increase over time was calculated Ex 485/Em 535 (5 h–0 h). Candidalysin treatment led to an unspecific fluorescence signal and was therefore excluded from analysis. All samples were prepared at least in duplicates.

**Western blot analysis**. LPS-primed (50 ng/ml, 2 h or 4 h) and unprimed (no LPS) hMDMs, mBMDMs or mBMDCs were seeded at $1.5 \times 10^6$ hMDMs or mBMDMs/well in 6-well (hMDMs, mBMDMs) or $1 \times 10^5$ mBMDCs/well in 96-well plates. Cells were infected with *C. albicans* (MOI 6 or 5), treated with synthetic Candidalysin, or 1 μM (hMDMs, mBMDMs)/5 μM (mBMDCs) Nigericin as a positive control. Supernatants were collected 5 h *p.i.*

For hMDMs and mBMDMs, supernatant proteins were precipitated with chloroform/methanol (1:4). The resulting protein pellets were resuspended in 1× Laemmli buffer (31.25 mM Tris-HCl pH 6.8, 12.5% glycerol, 1% SDS, 0.005% Bromophenol blue). For SDS-PAGE, 10 μL of supernatant sample (heat-denatured, with β-mercaptoethanol) were separated and transferred to a PVDF membrane.

Membranes were blocked with 5% bovine serum albumin (SERVA) in TBS-T (50 mM Tris, 0.15 M NaCl, 0.05% Tween 20, pH 7.6) and incubated with primary antibodies specific for IL-1β (AF-201-NA for human samples or AF-401-NA for murine samples; 1:800, R&D Systems) or caspase-1 (AG-20B-0048 for human samples or AG-20B-0042 for murine samples, 1:500, Adipogen) in TBS-T overnight at 4 °C. After washing three times with TBS-T, the membrane was incubated with horseradish peroxidase-conjugated anti-goat (HAF109, 1:2000, R&D Systems) or anti-mouse (1:2000, HAF007, R&D Systems) antibodies in TBS-T followed by three washing steps. Immunoreactivity was detected by enhanced chemiluminescence (ECL Plus western blotting Substrate; Thermo Fisher Scientific, Inc.). For mBMDCs, 15 μL of supernatant triplicate samples were pooled and boiled in 1× Laemmli buffer containing β-mercaptoethanol. Fifteen microliters of the pooled supernatant sample were separated by SDS-PAGE and transferred to a nitrocellulose membrane. Membranes were blocked and washed as described above and incubated with primary antibody specific for caspase-1 (AG-20B-0042, 1:1000, Adipogen). The membrane was washed as described above and incubated with a horseradish peroxidase-conjugated anti-mouse antibody (#7076, 1:2000, Cell Signaling). Ponceau or Coomassie staining of membrane or gel was used to ensure equal loading of supernatant samples to the gel. Full-size scans of western blots are provided in Supplementary Fig. 1.

**Statistical analysis**. Experiments were performed at least in biological triplicates ($n \geq 3$) with at least three different donors (hMDMs) or three independent experiments or mice (mBMDMs, mBMDCs, RAW264.7 Dectin-1), unless stated differently. Experiments performed in $Nlrp3^{-/-}$, $Pycard^{-/-}$ or $Casp1^{-/-}$ mBMDCs were performed in biological duplicates. All experiments were performed in an unblinded fashion. All data are reported as the scatterplot with median or for line charts mean + SD. No exclusion of data was performed except for Candidalysin induced TNF release, which was confirmed to be false positive in repeated experiments. Data were analysed using GraphPad Prism 7 (GraphPad Software, Inc. La Jolla, USA) and a one-way ANOVA for inter-group comparisons with a Dunnett's multiple comparison test. For statistical analysis of grouped data, a two-way ANOVA with Sidak's multiple comparison was applied. Statistically significant results are marked with a single asterisk meaning $p \leq 0.05$, double asterisks meaning $p \leq 0.01$ or triple asterisks meaning $p \leq 0.001$, nd—not detectable, n/a—not applicable.

## Data availability

The authors declare that the data supporting the findings of this study are available within the paper and its supplementary information files. All relevant data are available by request from the authors, with the restriction of data that would compromise the confidentiality of blood donors.

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

## Acknowledgements

We thank Ilse Jacobsen, Bianca Schulze, Katja Schubert, Silke Machata, and Marcel Sprenger for their help and support during isolation of mBMDMs; Stephanie Wisgott, Nadja Jablonowski, Daniel Fischer, Marcel Sprenger, Fabrice Hille, and Dorothee Eckhardt for their help and support during isolation and cultivation of hMDMs; Stephanie Wisgott additionally for technical assistance in western blotting and Tanja Neumayer and Valentin Höfl for technical assistance in mBMDC studies. Further, we thank Selene Mogavero for help during handling of synthetic peptides. The auto-MACS system for magnetic isolation of human monocytes was provided by the research group Fungal Septomics. This work was supported by the Deutsche Forschungsgemeinschaft SPP 1580 (Hu 528/17–1) to B.H. and L.K. and CRC/TR FungiNet Project C1 to B.H., as well as SFB 1054, SFB 1335 and RU 695/6–1 to J.R.; the Leibniz Association Campus InfectoOptics SAS-2015-HKI-LWC to B.H. and A.K.; the Bavarian Ministry of Sciences, Research and the Arts in the Framework of the Bavarian Molecular Biosystems Research Network (BioSysNet) to O.G.; a European Research Council (ERC) Advanced Grant (FP7, grant agreement no 322865) to J.R.; and an ERC Starting Grant (337689) to O.G.; EMBO Long-Term Fellowship (ALTF 18–2016) to J.W.; a Alexander von Humboldt postdoctoral research fellowship to M.S.G.; and grants from the Medical Research Council (MR/M011372/1), Biotechnology & Biological Sciences Research Council (BB/N014677/1), FP7-PEOPLE-2013-Initial Training Network (606786), National Institutes of Health (R37-DE022550), King's Health Partners Challenge Fund (R170501), the Rosetrees Trust (M680), and the NIH Research at Guys and St. Thomas's NHS Foundation Trust and the King's College London Biomedical Research Centre (IS-BRC-1215–20006) to J.R.N.; and the Division of Intramural Research, National Institute of Allergy and Infectious Diseases, NIH to R.D. and M.L.

## Author contributions

L.K., A.K., and P.A.K. contributed equally to this work. L.K. and A.K. performed hMDM and mBMDM experiments, analysed the data, wrote the manuscript and prepared the figures. P.A.K. performed mBMDC experiments, analysed the data, and edited the manuscript. M.S.G. performed hMDM and mBMDM experiments, analysed the data, and edited the manuscript. R.D. and M.S.L. performed in vivo experiments, analysed the data, and edited the manuscript. J.W. performed all experiments concerning RAW264.7 Dectin-1 macrophages. O.G. provided bone marrow of knock-out mice and edited the manuscript. J.R. and J.R.N. designed experiments and edited the manuscript. B.H. conceived and designed the study and wrote the manuscript.

## Additional information

**Competing interests:** The authors declare no competing interests.

