## [Peer Review File · Nature Communications]

Reviewers' Comments:

Reviewer #1:

Remarks to the Author:

This is a thorough and detailed experimental analysis of the role of the fungal toxin candidalysin during interaction with human and murine phagocytes. The authors demonstrate that CL acts as signal 2 in activation of the NLRP3 inflammasome, leading to production of IL-1b, which is important in resistance to systemic candidiasis and dissemination from the oral cavity as well as being involved in the immunopathogenesis of VVC. They have also identified K efflux as the mechanistic trigger involved. Further, they demonstrate that CL causes inflammasome independent cytolysis, revealing a third mechanism for phagocytic escape. Overall, this is an excellent manuscript, with appropriate controls and statistical analysis. There are several issues that need further discussion and clarification.

Major Points

- Candidalysin is required for virulence in mucosal infections but is not required for virulence during systemic infection. A more thorough and open discussion of the potential role for its effects on monocytes/DCs is warranted. Neither cells type plays a major role in VVC or oral candidiasis. What is the significance of these studies in relation to pathogenesis of any form of candidiasis? Dissemination from GI tract
- Differences in murine vs human cells:
- Line 489: candidalysin is sufficient but not necessary for IL-1b in murine cells. It seems the interpretation here should be that there are other fungal factor/s that exhibit redundancy in stimulating IL-1b as opposed to being required for FULL inflammasome activation. i.e. other factor/s compensate for loss of candidalysin as there was no reduction in IL-1b observed with *ece1* mutants.
- Is there any reason to suspect that the lack of phenotype for *ece1* mutants in systemic models of *C. albicans* infection is relate to lack of requirement for candidalysin in murine cells?
- NLRP3 activation and K influx: for consistency, it is necessary to show similar mechanism with murine cells

Reviewer #2:

Remarks to the Author:

The manuscript by Kasper et al focuses on the *Candida albicans* peptide toxin candidalysin, specifically examining its ability to trigger NLRP3-inflammasome activation and cell death in human and murine innate immune phagocytes.

Inflammasome activation by *C. albicans* in macrophages is an important host defence processes, because it enables the maturation and secretion of the major antifungal cytokine IL1beta. Somewhat paradoxically, this same process also allows *C. albicans* cells to escape from macrophages, due to caspase-1-dependent pyroptotic lysis of the host cell. How these processes are controlled by the pathogen and/or host to ultimately lead to either protection from infection (if the host wins) or disease (if the pathogen wins) is not clear. This is an important question in the field.

Little is understood about *Candida*-dependent processes that constitute the trigger for inflammasome activation. Based on our current understanding, the candidalysin-encoding gene *ECE1* is certainly a good candidate to be involved in inflammasome activation by *C. albicans* and induction of host cell lysis. The conclusions from the current manuscript are that candidalysin triggers NLRP3-caspase 1 inflammasome activation, leading to IL1b production. Candidalysin also triggers host cell lysis, however, the authors conclude that in this context, host cell death is inflammasome-independent.

Appropriately, the experiments to implicate candidalysin in inflammasome-dependent processes were performed either by using purified peptide (to isolate the effect of the toxin from other fungal factors), or by comparing wild type *C. albicans* to the *ece1* mutant (to address the importance of the toxin in the context of a live *Candida* infection of immune cells).

While the data using purified peptide is for the most part convincing, the *ece1* mutant data is not always convincingly showing an effect of candidalysin on the inflammasome-dependent phenotypes that were assessed. This is in my view a major issue with the current study. Using a substantial amount of purified peptide in *ex vivo* assays might not be fully relevant to an infection. Therefore, at the moment it is not clear if and to what extent ECE1/candidalysin is needed for inflammasome activation in the context of a live *Candida* immune cell challenge or during an *in vivo* infection. I also suggest that there is not enough evidence from human macrophage infections presented here to conclude that candidalysin-induced host cell lysis is inflammasome-independent. In fact, some of the presented data could suggest otherwise (I discuss this in more detail below).

Some specific comments that the authors might wish to consider:

1) While the *ece1* mutant was triggering less IL1b secretion by human MDMs as determined by ELISA (which does not discriminate mature IL1b from total, Fig 1B), Western blots to specifically assess mature IL1b and active casp1 (Fig 1C and 3D) did not show differences between wild type *Candida* and the *ece1* mutant. I note that the signal for active casp1 (p20) was reduced in *ece1* mutant infections (Fig 3D), but the pro-caspase was also reduced, so it is not clear if this is a loading issue. If ECE1 is needed to trigger inflammasome activation by *Candida* in human macrophages, then one would expect that the levels of active casp1 and mature IL1b will be reduced when macrophages are challenged by the mutant. An explanation is needed here.

In experiments where murine macrophages and DCs were used, the *ece1* mutant was not compromised in triggering IL1b secretion in ELISA assays, also showing that ECE1 might not be critical during live *Candida* challenge.

2) The effects of candidalysin on macrophage lysis in human macrophage infections (Figure 5C and D).

The live cell imaging of macrophage cell death in Fig 5C is interesting, and the biphasic kinetics of death of human macrophages is clear. The *ece1* mutant also clearly has an effect in this assay – with both phases of macrophage death affected, but there are some issues with the complemented strain as I elaborate below. Also, in my view, the mutant data here might support a role for ECE1 in pyroptosis, which is in contrast to the authors' conclusions based on and the data with purified candidalysin and mouse immune cells in Fig 6.

Based on Uwamahoro et al which used mouse BMDMs, the first wave of death is presumably inflammasome dependent, and the second is not. This assumption should be tested by the authors in their assay with human macrophages, by using a casp 1 inhibitor. This is important, because the first wave of death (which is likely casp1-dependent pyroptosis) is clearly reduced in the *ece1* mutant. This result would therefore support the idea that ECE1 is needed for activating inflammasome-dependent death of human macrophages (i.e. pyroptosis).

One problem with the data is that complementation of the mutant strain (*ece1*+ECE1) does not restore the first wave of death to wild type levels (this is particularly clear with donor 1 and donor 3) – this issue needs to be resolved. One option is to use multiple independent *ece1* mutant clones and ask if they all show reduced ability to lyse human macrophages, and also test multiple complemented clones. The effects of casp1 inhibitor on *ece1* mutant infection and macrophage lysis should also be tested. It would also be important to address if the fast death of human macrophages triggered by purified candidalysin (Fig 5D) is reduced by casp1 inhibition.

3) A final comment is related to in vivo infection – can the authors provide evidence that candidalysin/ECE1 is needed for inflammasome activation by *Candida* during in vivo infection? Does the *ece1* mutant trigger less IL1b in mouse infections? While ex vivo with BMDMs the *ece1* mutant behaved like wild type for IL1b secretion (Fig 2), maybe something would be revealed in vivo.

Reviewer #3:

Remarks to the Author:

Kasper et al. present a detailed in vitro characterization of the activities of the fungal peptide toxin Candidalysin in human and murine mononuclear phagocytes and identify different mechanisms, either dependent on or independent of Candidalysin, mediating the toxic effects of *C. albicans* on such cells. The mechanistic dissection is thorough, the experiments are well performed and logically presented. Although the in vitro characterization is extensive, translation to in vivo models of *C. albicans* infection is lacking. In addition, although the title clearly refers to mononuclear phagocytes, an exhaustive characterization of the effects on phagocytes, as I assume was the authors intention, would have benefit from an analysis of neutrophils. Beyond these general considerations, that do not question the validity of the manuscript, a major concern is that Candidalysin, while able to induce IL-1b production and caspase-1 when used to treat murine phagocytes, does not seem to play such a role during *C. albicans* infection (Figure 2A, 2B and 3D). Therefore, I find confusing sentences such as “additional unidentified fungal factors are required for full inflammasome activation in murine phagocytes” (lanes 489-490), or “Candidalysin is necessary for optimal inflammasome activation by *C. albicans* in [...] murine phagocytes” (lanes 661-662).

Minor concerns are detailed below:

Minor concerns:

- Figure 1B and throughout the manuscript: how were doses of Candidalysin selected for cell treatment? What are the amounts of Candidalysin secreted during *C. albicans* infection in the experimental models used?
- the caspase-1 inhibitor Z-YVAD-FMK seems to inhibit only partially IL-1b secretion in human MDMs upon infection with *C. albicans* or exposure to Candidalysin (Figure 3, A1) while the inhibition is almost complete in murine BMDMs. It seems to indicate that mechanisms involved are somewhat different.
- Figure 2C and 3F: both panels refer to unprimed murine BMDCs, 4h. Candidalysin induces TNF α in one panel (3F), but not in the other (2C). Is TNF α induced by Candidalysin in murine BMDCs?
- Figure 4E: lanes 665-667 “In agreement with this, the *C. albicans* strains *efg1* Δ/Δ *cph1* Δ/Δ and *hcg1* Δ/Δ [...] induced no or very little IL-1b secretion”; the panel shows similar values between these strains and *ece1* Δ/Δ
- lanes 684-685: “we demonstrate that Candidalysin is required for the full damage potential of *C. albicans*”. As reported by the authors, the difference between WT and *ece1* Δ/Δ +ECE1 Δ 184-279 is not significant, so the sentence should be modified
- lanes 786-787: “*C. albicans*-induced mBMDM damage required LPS-priming and was reduced after caspase-1 inhibition (Figure 6C)”: are the differences statistically significant?

Minor (technical)

- panel 1E: font size is too low and very difficult to read; moreover, the third column of each graph of this panel should refer to *ece1* Δ/Δ +ECE1 Δ 184-279 according to the main text
- panel organization in some figures is questionable. For instance, Figure 1 is related to human MDMs with the exception of panel D, which refers to murine BMDMs. Murine BMDMs are addressed in Figure 2 and, in my opinion, panel D is more appropriate in this Figure.
- Suppl Figure 1C: could the authors provide statistical significance between untreated and Candidalysin?
- Figure 3D (human MDMs): caspase 1 p20 is difficult to assess.

NCOMMS-18-07336 Kasper *et al.* “The fungal peptide toxin Candidalysin activates the NLRP3 inflammasome and causes cytolysis in mononuclear phagocytes”

Point to point replies

General comments:

To investigate the role of Candidalysin for macrophage inflammasome activation and killing, we used both, synthetic peptides and *C. albicans* mutant strains and two different types of macrophages from humans (hMDMs) and mice (mBMDMs) and dendritic cells (mBMDCs) from mice and investigated both, inflammasome activation and host cell killing on multiple levels.

All three reviewers acknowledged and did not challenge our main statements: that the fungal peptide toxin Candidalysin activates the NLRP3 inflammasome and causes cytolysis in mononuclear phagocytes.

Reviewer 1 stressed that differences are seen between human and murine cells, and acknowledged the study in general and the fact that appropriate controls and statistical analysis had been done. Reviewer 1 requested further discussion and clarification and one set of experiments for completeness.

The main criticisms of reviewer 2 and 3 focus on the differences between macrophages from humans and macrophages/DCs from mice and differences between synthetic Candidalysin and *C. albicans* mutant strains, and both reviewers requested *in vivo* evidence for Candidalysin-mediated IL-1 β induction.

We have dealt with all critical points, have included additional data, repeated experiments, performed additional experiments, including *in vivo* infections and quantification of bioactive IL-1 β , and modified specific conclusions when necessary.

We further have shortened parts of the discussion and extended other parts in response to the reviewers and new published information related to the study.

All bar graphs are now shown as scatter dot plots according the journal guidelines.

Reviewer #1 (Remarks to the Author):

This is a thorough and detailed experimental analysis of the role of the fungal toxin candidalysin during interaction with human and murine phagocytes. The authors demonstrate that CL acts as signal 2 in activation of the NLRP3 inflammasome, leading to production of IL-1b, which is important in resistance to systemic candidiasis and dissemination from the oral cavity as well as being involved in the immunopathogenesis of VVC. They have also identified K efflux as the mechanistic trigger involved. Further, they demonstrate that CL causes inflammasome independent cytolysis, revealing a third mechanism for phagocytic escape. Overall, this is an excellent manuscript, with appropriate controls and statistical analysis. There are several issues that need further discussion and clarification.

Reply:

We thank Reviewer 1 for these encouraging comments.

Major Points

- Candidalysin is required for virulence in mucosal infections but is not required for virulence during systemic infection. A more thorough and open discussion of the potential role for its effects on monocytes/DCs is warranted. Neither cells type plays a major role in VVC or oral candidiasis. What is the significance of these studies in relation to pathogenesis of any form of candidiasis?

Reply:

Our published and unpublished data clearly point towards (a) dual roles of Candidalysin in *C. albicans* pathogenesis and (b) different roles of Candidalysin in different types of infection: (a) Clearly Candidalysin is both, a classical virulence factor causing host cell damage (Casadevall and Pirofski, 2014) and an avirulence factor or immunomodulator (White *et al.*, 2000; Medzhitov *et al.*, 2012), which is recognized by the host immune system to initiate a protective response. The outcome of the two effects, damage potential versus protective immune response, dictates the outcome of the infection. (b) During oral infections, epithelial cells recognize Candidalysin *via* the danger response pathway (*via* p38 and c-Fos) (Moyes *et al.* 2010), causing cytokine production and ultimately attraction of phagocytes, in particular neutrophils. This attraction of neutrophils is a key event, but with oppositional outcome in different tissues (and depending on the immune status of the host). During oral infections, attraction of neutrophils is protective in immunocompetent mice (Verma *et al.* 2017), while attraction of neutrophils in vaginal infections is associated with immunopathology (Richardson *et al.* 2018). We believe that similar processes occur in tissues during systemic infections with macrophages being key players responsible for neutrophil attraction. This concept is, for example, in agreement with the observation that organ specific fungal morphology and neutrophil attraction correlates with pathogenesis (Lionakis *et al.* 2011).

This has now been discussed in the revised manuscript (**lines 557-574 in the manuscript with highlighted changes**).

- Differences in murine vs human cells:
- Line 489: candidalysin is sufficient but not necessary for IL-1b in murine cells. It seems the interpretation here should be that there are other fungal factor/s that exhibit redundancy in stimulating IL-1b as opposed to being required for FULL inflammasome activation. i.e. other factor/s compensate for loss of candidalysin as there was no reduction in IL-1b observed with *ece1* mutants.

Reply:

We thank Reviewer 1 for pointing this out and providing a more precise statement (see also Reviewer 3). A similar statement has now been added to the revised manuscript. We also stressed that Candidalysin is obviously more important for human cells as compared to murine cells. (**lines 211-213 and 451-474**)

- Is there any reason to suspect that the lack of phenotype for *ece1* mutants in systemic models of *C. albicans* infection is relate to lack of requirement for candidalysin in murine cells?

Reply:

The reviewer is correct that, based on our observation that the *ece1* Δ/Δ mutant and the *ece1* Δ/Δ +*ECE1* _{Δ 184-279} do not induce reduced levels of IL-1 β in murine BMDMs and murine BMDCs,

one may anticipate that a lack of Candidalysin expression in these mutants would translate into absence of a phenotype (reduced IL-1 β levels) *in vivo*. In contrast, the Candidalysin peptide shows, similar to human cells, a strong IL-1 β induction in murine cells, which highlights that Candidalysin can have strong effects on murine cells.

The role of Candidalysin/Ece1 during systemic infections has, to our knowledge, not been published yet. However, preliminary data of our collaboration partners suggest that Candidalysin does play a role during systemic infections in mice depending on the immune status (manuscripts in preparation), which is in agreement with the dual role of Candidalysin discussed above and in the Discussion section (**lines 558-575**) of the revised manuscript.

Moreover, as also requested by reviewers 2 and 3, we have performed an additional *in vivo* experiment (in collaboration with the team of Mihail Lionakis) in which mice were systemically infected with wild type *C. albicans* (BWP17) or the *ece1* Δ/Δ mutant. At 1 day *post infection* IL-1 β was measured in kidney and spleen homogenates. These two organs were chosen, because Lionakis *et al.* (2011) showed that the morphology of infecting *C. albicans* cells differs between these organs. Thus, as expression of Candidalysin (*ECE1*) is restricted to hyphae, we expected only differences between the wild type and the *ece1* Δ mutant in organs where hyphal morphology is observed. In fact, in the kidney, where the fungal morphology is dominated by hyphae (Lionakis *et al.* 2011), IL-1 β levels were significantly lower in mice infected with the *ece1* Δ/Δ knockout strain compared to mice infected with wild type *C. albicans* cells. Whereas in the spleen, where the prevalent fungal morphology is yeasts (Lionakis *et al.* 2011), morphology is dominated by yeasts (Lionakis *et al.* 2011), no differences in IL-1 β levels were observed. This suggests that Candidalysin is in fact required for full inflammasome activation *in vivo* in mice. We included these new data as **Figure 1** in the revised manuscript.

It should also be noted that we observed a clearer, although not statistically significant, trend towards a reduced capacity of the *ece1* Δ/Δ mutants to induce IL-1 β secretion (resembling the pattern observed with hMDMs) when we repeated experiments with mBMDCs to clarify whether Candidalysin-induced IL-1 β secretion is caspase-1 and NLRP3 inflammasome-dependent (**Figure 4f**).

- NLRP3 activation and K influx: for consistency, it is necessary to show similar mechanism with murine cells

Reply:

We agree and have performed additional experiments to confirm that potassium efflux plays a significant role in inflammasome activation by Candidalysin in mBMDMs and mBMDCs similar to Fig. 4a (addition of KCl and Glibenclamide). Our new data, similar to our results derived from human MDMs, indicate that potassium efflux is also required for Candidalysin-induced inflammasome activation in murine BMDMs (see new **Figure 5B** in the revised manuscript) and BMDCs (see new **Figure 5C** in the revised manuscript), as addition of exogenous potassium in the form of KCl reduced IL-1 β release. Furthermore, addition of Glibenclamide led to reduced IL-1 β levels in mBMDMs and mBMDCs. TNF levels were not altered in a corresponding control experiment.

LPS-primed murine BMDMs 5h

LPS-primed murine BMDCs, 4h

Reviewer #2 (Remarks to the Author):

The manuscript by Kasper et al focuses on the *Candida albicans* peptide toxin candidalysin, specifically examining its ability to trigger NLRP3-inflammasome activation and cell death in human and murine innate immune phagocytes.

Inflammasome activation by *C. albicans* in macrophages is an important host defence processes, because it enables the maturation and secretion of the major antifungal cytokine IL1 β . Somewhat paradoxically, this same process also allows *C. albicans* cells to escape from macrophages, due to caspase-1-dependent pyroptotic lysis of the host cell. How these processes are controlled by the pathogen and/or host to ultimately lead to either protection from infection (if the host wins) or disease (if the pathogen wins) is not clear. This is an important question in the field.

Little is understood about Candida-dependent processes that constitute the trigger for inflammasome activation. Based on our current understanding, the candidalysin-encoding gene ECE1 is certainly a good candidate to be involved in inflammasome activation by *C. albicans* and induction of host cell lysis. The conclusions from the current manuscript are that candidalysin triggers NLRP3-caspase 1 inflammasome activation, leading to IL1b production. Candidalysin also triggers host cell lysis, however, the authors conclude that in this context, host cell death is inflammasome-independent.

Appropriately, the experiments to implicate candidalysin in inflammasome-dependent processes were performed either by using purified peptide (to isolate the effect of the toxin from other fungal factors), or by comparing wild type *C. albicans* to the *ece1* mutant (to address the importance of the toxin in the context of a live Candida infection of immune cells).

While the data using purified peptide is for the most part convincing, the *ece1* mutant data is not always convincingly showing an effect of candidalysin on the inflammasome-dependent phenotypes that were assessed. This is in my view a major issue with the current study. Using a substantial amount of purified peptide in ex vivo assays might not be fully relevant to an infection.

Reply:

We agree that data with synthetic peptides provide a much clearer picture of the role of Candidalysin in cell death and inflammasome activation as this allows “to isolate the effect of the toxin from other fungal factors”, as correctly stated by Reviewer 2. In fact, we believe that the effect of Candidalysin can be both, intensified in the presence of fungal cells (in particular hypha), but also covered by other factors of fungal cells, which are also relevant for host recognition and inflammasome activation. The latter seems to be the case in this study, depending on the origin of the phagocytes. We discuss this aspect in **lines 211-213 and 451-474 (in the manuscript with highlighted changes)** of the revised manuscript.

W.r.t. peptide concentrations: In our studies we have used lytic and sub-lytic concentrations of Candidalysin and even sub-lytic concentrations are sufficient to activate the danger response pathway in epithelial cells. In Moyes *et al.* (2016) we proposed a model of *C. albicans* mucosal infection whereby invasive hyphae secrete Candidalysin into a membrane-bound ‘invasion pocket’, facilitating peptide accumulation. During early stages of infection, sub-lytic concentrations of Candidalysin induce epithelial immunity by activating the “danger response” pathway, alerting the host to the transition from colonizing yeast to invasive, toxin-producing, hyphae. Similarly, we propose that high local concentrations of Candidalysin can be reached within macrophages phagosomes or local inflammatory lesions within organs with high numbers of hyphae.

Therefore, at the moment it is not clear if and to what extent ECE1/candidalysin is needed for inflammasome activation in the context of a live Candida immune cell challenge or during an in vivo infection.

Reply:

To investigate the extent of the role of *ece1Δ* in IL-1β induction, we now have included *in vivo* data showing increased IL-1β production in wild type versus *ece1Δ* mutant infected kidneys of mice (see also response to reviewer 1 and 3). At 1 day *post infection* IL-1β was measured in kidney and spleen homogenates. These two organs were chosen, because Lionakis *et al.* (2011) showed that the

morphology of infecting *C. albicans* cells differs between these organs. Thus, as expression of Candidalysin (*ECE1*) is restricted to hyphae, we expected only differences between the wild type and the *ece1Δ* mutant in organs where hyphal morphology is observed. In fact, in the kidney, where the fungal morphology is dominated by hyphae (Lionakis *et al.* 2011), IL-1 β levels were significantly lower in mice infected with the *ece1Δ/Δ* knockout compared to Wt infected mice. Whereas in the spleen, where the fungal morphology is dominated by yeasts (Lionakis *et al.* 2011), no differences in IL-1 β levels were observed. We included these new data as **Figure 1** of the revised manuscript (see also response to reviewer 1).

We believe that this together with our *in vitro* data highlights the relevance of our study for *in vivo* infections.

It should be noted that our observations point to a more important role of Candidalysin in human macrophage inflammasome activation as compared to mice within the context of fungal cells rather than the Candidalysin peptide alone. Our additional experiments using murine BMDMs and BMDCs illustrate that the mechanism of Candidalysin-induced inflammasome activation relies both in human as well as in murine cells on potassium efflux (see new **Figure 5B and C** in the revised manuscript; see also response to reviewer 1.).

A primary reason is the fact that with synthetic peptides the observed responses are solely due to the added peptide and not clouded by other fungal molecules. Apart from Candidalysin, *C. albicans* has a whole collection of molecules that are immunogenic and can elicit, IL-1 β dominated, immune responses. Thus, as correctly stated by reviewer 1: “there are other fungal factor/s that exhibit redundancy in stimulating IL-1 β as opposed to being required for full inflammasome activation”. This has been discussed in the revised manuscript (**lines 211-213 and 451-474**).

I also suggest that there is not enough evidence from human macrophage infections presented here to conclude that candidalysin-induced host cell lysis is inflammasome-independent. In fact, some of the presented data could suggest otherwise (I discuss this in more detail below).

Some specific comments that the authors might wish to consider:

1) While the *ece1* mutant was triggering less IL1 β secretion by human MDMs as determined by ELISA (which does not discriminate mature IL1 β from total, Fig 1B), Western blots to specifically assess mature IL1 β and active casp1 (Fig 1C and 3D) did not show differences between wild type *Candida* and the *ece1* mutant. I note that the signal for active casp1 (p20) was reduced in *ece1* mutant infections (Fig 3D), but the pro-caspase was also reduced, so it is not clear if this is a loading issue. If *ECE1* is needed to trigger inflammasome activation by *Candida* in human macrophages, then one would expect that the levels of active casp1 and mature IL1 β will be reduced when macrophages are challenged by the mutant. An explanation is needed here.

Reply:

Our IL-1 β assays in human MDMs (**Figure 2B** in the revised MS) clearly indicate that the *ece1Δ* mutants cause significantly less IL-1 β secretion by human MDMs as quantified by ELISA. Moreover, synthetic Candidalysin dose-dependently triggers strong production of IL-1 β . We continued to use western blotting to qualitatively show that mature IL-1 β is in fact produced. These data clearly show that synthetic Candidalysin alone triggers strong production of mature IL-1 β and that this requires priming/a first signal. Further, using murine BMDCs, the same trend as seen in the ELISA can also be

observed (less caspase-1 activation in the *ece1* Δ mutants; see **Figure 4E**). However, we agree that pro-caspase-1 levels were also reduced in murine BMDCs exposed to the *ece1* Δ mutants and that both the *ece1* Δ mutants and the wild type strain appear to induce similar amounts of secreted IL-1 β in mBMDCs.

We have repeated western blots a number of times and found it very difficult to get reliable quantitative data for IL-1 β secretion by human MDMs and for caspase-1 activation. Therefore, we decided to use a bioassay to quantify the amount of extracellular bioactive IL-1 β levels. The EL4.NOB-1 cell line has a high level of IL-1 receptor expression and reacts to bioactive IL-1 β by producing IL-2, but not to pro-IL-1 β . The culture supernatants of hMDMs and mBMDCs were used to stimulate the EL4.NOB-1 cell line and were compared to a standard curve of recombinant human IL-1 β . Based on the standard curve we estimated the levels of bioactive IL-1 β in the culture supernatants. Our data demonstrates that the synthetic Candidalysin peptide induces the release of significant amounts of bioactive IL-1 β in human and murine cells. Similarly we observe that wild type *C. albicans* cells induce significantly more bioactive IL-1 β release than the *ece1* Δ mutants in hMDMs. In mBMDCs we observed a trend towards reduced IL-1 β production by *ece1* Δ mutants as compared to the wild type, although this was not statistically significant. This is discussed below. These new data are now included as **Figure 2D and 3B** of the revised manuscript.

D LPS-primed human MDMs, 5 h

B LPS-primed murine BMDMs, 5 h

Similarly, by using a Caspase-GLO assay we demonstrate caspase-1 activity in mBMDCs stimulated with wild type *C. albicans*, but significantly reduced caspase-1 activity in mBMDCs exposed to the *ece1* Δ/Δ and *ece1* Δ/Δ +*ECE1* $\Delta_{184-279}$ mutants. As expected, caspase-1 activation by all strains was further reduced by addition of the caspase-1 inhibitor Ac-YVAD-CHO, indicating residual activation of caspase-1 by factors other than Candidalysin. These new data are now included as **Figure 4D** of the revised manuscript.

We believe that our new quantitative data provide convincing evidence for Candidalysin-dependent production of bioactive IL-1 β and caspase-1 activation. We have left the western blots in the figures of the revised manuscript as we believe that they contain important information.

In experiments where murine macrophages and DCs were used, the *ece1* mutant was not compromised in triggering IL1b secretion in ELISA assays, also showing that ECE1 might not be critical during live *Candida* challenge.

Reply:

Although the effect with synthetic Candidalysin was clear, we agree that, in contrast to LPS-primed human macrophages (Figure 2B,D in the revised manuscript), the effect with LPS-primed murine mononuclear phagocytes was less clear when *C. albicans ece1Δ/Δ* mutant strains were compared to the wild type (Figure 3A,B,D in the revised manuscript). However, according to the literature (Cullen *et al.* 2015), LPS alone induces necrotic cell death and IL-1 β production in murine cells and it seems clear that additional fungal factors may compensate for the loss of Candidalysin to trigger inflammasome activation in murine mononuclear phagocytes (in contrast to human macrophages). We have further clarified this in the Discussion section and have adjusted our conclusions in this part of the revised manuscript (lines 451-474). Again, we wish to stress that these facts suggest that Candidalysin seems to be more important for inflammasome activation in human macrophages as compared to the murine model host. Nevertheless, our *in vivo* data (see response to reviewer 1 and Figure 1 of the revised manuscript) highlight a role for Candidalysin in inducing IL-1 β during systemic *C. albicans* infection in mice and infection of one of the crucial target organs, the kidney.

2) The effects of candidalysin on macrophage lysis in human macrophage infections (Figure 5C and D).

The live cell imaging of macrophage cell death in Fig 5C is interesting, and the biphasic kinetics of death of human macrophages is clear. The *ece1* mutant also clearly has an effect in this assay – with both phases of macrophage death affected, but there are some issues with the complemented strain as I elaborate below. Also, in my view, the mutant data here might support a role for ECE1 in pyroptosis, which is in contrast to the authors' conclusions based on and the data with purified candidalysin and mouse immune cells in Fig 6.

Reply:

We agree that the *ece1Δ/Δ* mutant clearly has attenuated abilities to damage macrophages and that the cell death dynamics (presented in **Figure 6** of the revised manuscript) may suggest a minor contribution of Candidalysin to the pyroptotic phase based on the bi-phasic dynamics of host cell damage. This could be either a contribution to pyroptosis or a non-pyroptotic contribution to this predominantly pyroptotic phase. However, experiments with murine inflammasome knock out cells (shown in **Figure 7** of the revised manuscript) clearly show that cell damage by Candidalysin in murine macrophages is largely independent of caspase-1 and the NLRP3 inflammasome and thus not due to pyroptosis. However, a minor contribution seems possible or, in the context of live *C. albicans*, Candidalysin might facilitate induction of pyroptosis by other fungal molecules. We have thus discussed this in the manuscript accordingly (**lines 495-507** in the revised manuscript.).

Based on Uwamahoro et al which used mouse BMDMs, the first wave of death is presumably inflammasome dependent, and the second is not. This assumption should be tested by the authors in their assay with human macrophages, by using a casp 1 inhibitor. This is important, because the first wave of death (which is likely casp1-dependent pyroptosis) is clearly reduced in the *ece1* mutant. This result would therefore support the idea that ECE1 is needed for activating inflammasome-dependent death of human macrophages (i.e. pyroptosis).

Reply:

We agree with Reviewer 2 that the pioneering work of Uwamahoro *et al.* (2014) suggests that, during interaction of *C. albicans* and murine macrophages, the first wave of death is presumably and predominantly inflammasome-dependent, and the second one is not. Our live cell imaging with human macrophages basically confirmed the cell damage dynamics observed by Uwamahoro *et al.* (2014) with moderate donor-specific variations. In our series of experiments to dissect the relation between Candidalysin-mediated cell death and the inflammasome, we also tested the effect of the caspase-1 inhibitor Z-YVAD-FMK on host cell damage (mBMDMs and hMDMs) caused by Candidalysin, by measuring LDH release 5 h post infection. Clearly, the caspase-1 inhibitor had no significant impact on Candidalysin induced cell death. These data are now included as **Figure 7 A,C** in the revised manuscript. Similarly, we performed additional experiments where we investigated the dynamics of cell death by the Candidalysin peptide in the presence or absence of the caspase-1 inhibitor VX675 (we included these new data as **Figure 7E** in the revised manuscript). Again we did not observe any significant effect on Candidalysin-induced cell death. Collectively we conclude from these data that the Candidalysin peptide by itself does not induce pyroptotic cell death.

E Candidalysin-induced cell death
LPS-primed human MDMs

In the context of *C. albicans* wild type and *ece1Δ* mutant cells the case is less clear. As correctly observed by reviewer 2, the first wave of death is clearly reduced in the *ece1Δ* mutant, which could hint towards reduced pyroptosis (or a non-pyroptotic contribution to the predominantly pyroptotic phase – see above). Inhibition of Caspase-1 using Z-YVAD-FMK caused moderately, but non-significantly, reduced LDH release in mBMDMs after 5h (see **Figure 7B** in the revised manuscript). We now included additional data using human MDMs and murine BMDCs and caspase-1 inhibitors, however, no significant effect of caspase-1 inhibition was observed on LDH release from wild type cells (included as **Figure 7A,C** in the revised manuscript), except for a minor non-significant reduction in mBMDCs.

We also have tried to further elucidate this aspect using the dynamic live cell imaging of macrophage cell death for human cells using the Z-YVAD-FMK caspase-1 inhibitor, but have not observed any reduction of cell death due to the inhibitor. Rather the PI signal was increased.

Therefore, this inhibitor seems to be inappropriate for this type of experiment. We therefore tested an alternative caspase-1 inhibitor VX675 (also used for the Candidalysin induced cell death dynamics for this reason). Using VX675 we were unable to identify any effect of caspase-1 on the *C. albicans*-induced cell death (we included these new data as **Figure 7D** in the revised manuscript).

D

Candida albicans-induced cell death
LPS-primed human MDMs

To ultimately prove whether the cause of macrophage cell damage by Candidalysin depends on the inflammasome and is pyroptotic or non-pyroptotic, we chose a genetic approach and have used knock out mice strains. These experiments (shown in **Figure 7G** of the revised manuscript) clearly show that Candidalysin-induced cell damage in murine macrophages is completely independent of caspase-1 and the NLRP3 inflammasome and thus not due to pyroptosis.

Nevertheless, confirming published data, a large portion of cell damage in murine macrophages induced by *C. albicans* cells is dependent of caspase-1 and the NLRP3 inflammasome and thus due to pyroptosis. Although the level of LDH release in inflammasome knock out phagocytes is similar to wild type phagocytes exposed to the *ece1Δ* mutant, the inflammasome knockout phagocytes have even further reduced LDH release indicating that pyroptosis plays a major role in *C. albicans*-induced cell death by Candidalysin deficient cells.

This could suggest that the first wave of death observed in macrophages exposed to wild type *C. albicans* cells (Figure 6) might be the additive effect of Candidalysin dependent mechanisms and pyroptosis dependent mechanisms, whereas the first wave of death observed in macrophages exposed to the *ece1Δ* mutant may be predominantly pyroptosis dependent. Combining the data obtained from murine inflammasome knockout BMDMs and human MDMs we are confident that Candidalysin itself does not significantly contribute to pyroptosis. However, it is not possible to exclude that Candidalysin might facilitate the induction of pyroptosis by other fungal molecules present in the context of a fungal cell. This has now been added to the Discussion section of the revised manuscript (**lines 495-507**).

In summary, we conclude that the first wave of damage caused by wild type *C. albicans* to human MDMs as seen in our experimental system is mainly due to non-pyroptotic mechanisms.

One problem with the data is that complementation of the mutant strain (*ece1+ECE1*) does not restore the first wave of death to wild type levels (this is particularly clear with donor 1 and donor 3) – this issue needs to be resolved. One option is to use multiple independent *ece1* mutant clones and ask if they all show reduced ability to lyse human macrophages, and also test multiple complemented clones. The effects of casp1 inhibitor on *ece1* mutant infection and macrophage lysis should also be tested. It would also be important to address if the fast death of human macrophages triggered by purified candidalysin (Fig 5D) is reduced by casp1 inhibition.

Reply:

Although the *ece1Δ+ECE1* revertant strain showed the same trend in all three experiments with three independent donors, we agree with Reviewer 2 that the *ece1Δ + ECE1* revertant does not fully restore the wild type damaging capacity. Instead the revertant showed an intermediate recovery of the wild type damaging capacity. However, such an intermediate phenotype recovery of revertant strains is frequently observed in *C. albicans* due to the gene dosage effect (one *ECE1* copy in the revertant as opposed to two copies in the wild type) and generally accepted as being sufficient to prove the case. The reliability of the revertant (including the comparison of independent revertant clones) has already been tested in the context of our original study dealing with the discovery of Candidalysin (Moyes *et al.* 2016). The revertant has also been successfully used in further studies (Richardson *et al.*, 2018a; Varma *et al.*, 2017; Richardson *et al.*, 2018b; Varma *et al.*, 2018; Allert, Foerster *et al.*, 2018).

3) A final comment is related to *in vivo* infection – can the authors provide evidence that candidalysin/ECE1 is needed for inflammasome activation by *Candida* during *in vivo* infection? Does the *ece1* mutant trigger less IL1b in mouse infections? While *ex vivo* with BMDMs the *ece1* mutant behaved like wild type for IL1b secretion (Fig 2), maybe something would be revealed *in vivo*.

Reply:

As discussed above and requested by reviewer 2 and 3, we performed an additional *in vivo* experiment in which mice were systemically infected with wild type *C. albicans* or the *ece1Δ* mutant. These data show significantly lower IL-1 β levels in kidneys (predominantly hyphae, see: Lionakis *et al.* (2011)) infected with the *ece1Δ* knockout as compared to the wild type. In contrast, no differences were observed in the spleen (pre-dominantly yeast cells). We included these new data as **Figure 1** in the revised manuscript (see response to reviewer 1 and 2).

Reviewer #3 (Remarks to the Author):

Kasper *et al.* present a detailed *in vitro* characterization of the activities of the fungal peptide toxin Candidalysin in human and murine mononuclear phagocytes and identify different mechanisms, either dependent on or independent of Candidalysin, mediating the toxic effects of *C. albicans* on such cells. The mechanistic dissection is thorough, the experiments are well performed and logically presented. Although the *in vitro* characterization is extensive, translation to *in vivo* models of *C. albicans* infection is lacking.

Reply:

As discussed above (reviewer 1 and 2), we have performed an additional *in vivo* experiment to translate our findings to an *in vivo* situation. Our data demonstrate that during systemic *C. albicans* infections, IL-1 β levels in the kidney, where the fungal morphology is dominated by hyphae (Lionakis *et al.* 2011), are significantly higher in mice infected with wild type *C. albicans* cells compared to the *ece1* Δ mutant that does not produce Candidalysin. Whereas in the spleen, where the fungal morphology is dominated by yeasts (Lionakis *et al.* 2011), no differences in IL-1 β levels were observed. We included these new data as **Figure 1** in the revised manuscript (see response to reviewer 1 and 2).

In addition, although the title clearly refers to mononuclear phagocytes, an exhaustive characterization of the effects on phagocytes, as I assume was the authors intention, would have benefit from an analysis of neutrophils.

Reply:

We appreciate the reviewer's suggestion; however, the main scope of our manuscript is the role of Candidalysin in the processing and release of IL-1 β . Neutrophils are not a major source of IL-1 β , have been reported to produce IL-1 β by non-inflammatory mechanisms, and, to our knowledge, have not been described to release IL-1 β in response to *C. albicans* stimulation. In fact, we have not been able to measure IL-1 β in supernatants of PMNs stimulated with *C. albicans* (Gresnigt *et al.* 2012 and unpublished data). Nevertheless, neutrophil antifungal activities through formation of Neutrophil Extracellular Traps (NETs) are also crucial for the host defense against fungi. The investigation of whether Candidalysin plays a role for in the induction of NET formation is, however, beyond the scope of the current manuscript. Though we can inform that studies regarding the effects of Candidalysin on NET formation are currently ongoing together with one of our collaborators.

Beyond these general considerations, that do not question the validity of the manuscript, a major concern is that Candidalysin, while able to induce IL-1 β production and caspase-1 when used to treat murine phagocytes, does not seem to play such a role during *C. albicans* infection (Figure 2A, 2B and 3D).

Reply:

As discussed above, we agree that the effects are clear with human macrophages, but less clear with murine phagocytes when using *C. albicans* cells. Indeed in **Figure 3 A, B and D** we do not see a clear phenotype of the Candidalysin-deficient *C. albicans* strains. Nonetheless, it should be noted that the Candidalysin peptide shows IL-1 β production and inflammasome activation consistently in murine BMDMs and BMDCs similar as in human macrophages. Furthermore, as suggested by reviewer 1 we

performed additional experiments that highlight that the inflammasome activation by Candidalysin in murine phagocytes is mechanistically mediated through potassium efflux similar to our findings in the original manuscript (these new data are included in **Figure 5 B,C** in the revised manuscript, see response to reviewer 1). We believe that the use of synthetic peptides is the ‘cleanest’ method of deciphering how Candidalysin shapes the immune response and mediates inflammasome activation. A primary reason is the fact that the observed responses to synthetic peptides are solely due to the added peptide and not clouded by other fungal molecules. Apart from Candidalysin, *C. albicans* has a whole collection of molecules that are immunogenic and can elicit, IL-1 β -dominated, immune responses. Thus, as correctly stated by reviewer 1: “there are other fungal factor/s that exhibit redundancy in stimulating IL-1 β as opposed to being required for full inflammasome activation”. This has been discussed in the revised manuscript (**lines 211-213 and 451-474 in the manuscript with highlighted changes**).

Why these other fungal factors affect murine BMDMs and BMDCs more than human MDMs may have several reasons. For example, the expression of PRRs that recognize fungal IL-1 β -inducing PAMPs might be different between human and murine cells. It has been shown that the immune response to other fungal pathogens, such as *Aspergillus fumigatus*, clearly differ between murine and human macrophages (Hellmann *et al.* 2017).

Despite the fact that we do not observe a clear reduction of IL-1 β levels in the *ece1* Δ mutant in murine macrophages and DCs, we are convinced that the inflammasome activation by Candidalysin is universal between mice and humans and now confirmed this by our new *in vivo* data (**Figure 1** of the revised manuscript, see response to reviewer 1 and 2). These show clear differences in IL-1 β release by wild type and *ece1* Δ cells in kidneys where *C. albicans* is predominantly found in its hyphal form, while no difference were observed in the spleen where the fungus is predominantly in the yeast form. These show clear differences in IL-1 β release by wild type and *ece1* Δ cells in kidneys, where *C. albicans* is predominantly found in its hyphal form, while no difference were observed in the spleen, where the fungus predominantly exists in the yeast form.

Minor concerns are detailed below:

Minor concerns:

- Figure 1B and throughout the manuscript: how were doses of Candidalysin selected for cell treatment? What are the amounts of Candidalysin secreted during *C. albicans* infection in the experimental models used?

Reply:

In our studies we have used lytic and sub-lytic concentrations of Candidalysin and even sub-lytic concentrations are sufficient to activate the danger response pathway in epithelial cells. In Moyes *et al.* (2016) we proposed a model of *C. albicans* mucosal infection whereby invasive hyphae secrete Candidalysin into a membrane-bound ‘invasion pocket’, facilitating peptide accumulation. During early stages of infection, sub-lytic concentrations of Candidalysin induce epithelial immunity by activating the “danger response” pathway, alerting the host to the transition from colonizing yeast to invasive, toxin-producing hyphae. Similarly, we propose that high concentrations of Candidalysin can be reached within macrophages phagosomes or local inflammatory lesions within organs with high numbers of hyphae (see also response to reviewer 2).

It should be noted that it is technically almost impossible to quantify concentrations of the Candidalysin peptide during interaction with host cells, in particular in the synapse of the host-pathogen interface, since the peptide seems to integrate very quickly into membranes

- the caspase-1 inhibitor Z-YVAD-FMK seems to inhibit only partially IL-1b secretion in human MDMs upon infection with *C. albicans* or exposure to Candidalysin (Figure 3, A1) while the inhibition is almost complete in murine BMDMs. It seems to indicate that mechanisms involved are somewhat different.

Reply:

The fact that the inhibitor only partially reduces IL-1 β secretion in human MDMs may be explained by several reasons. First, it must be noted that this inhibitor is dissolved in DMSO and the solvent control (DMSO alone) caused an increase in IL-1 β release in human macrophages (compared to the “no DMSO” control), but not in mouse cells, suggesting difference in the susceptibility of human and mouse cells to the used DMSO concentrations. Second, inhibitors like Z-YVAD-FMK that require specific binding to a binding pocket only on rare occasions can fully block effects due to high reaction constants of the binding to the target. Whereas, for example, extracellular addition of potassium chloride, which does not require binding to a specific target, fully blocks the Candidalysin-mediated inflammasome activation in both, human and murine cells (Figure 5 B,C of the revised manuscript, see response to reviewer 1).

We believe that the observed discrepancy in full blockade of IL-1 β release using the Z-YVAD-FMK inhibitor is rather due to technical differences of the *in vitro* system and the use of inhibitors rather than due to the involvement of other mechanisms.

- Figure 2C and 3F: both panels refer to unprimed murine BMDCs, 4h. Candidalysin induces TNF α in one panel (3F), but not in the other (2C). Is TNF α induced by Candidalysin in murine BMDCs?

Reply:

The reviewer correctly observed that there is some induction of TNF in the experiment with the knock out murine BMDCs. This is a single observation where we saw induction of TNF by Candidalysin. Therefore, we have repeated several experiments and re-measured TNF levels. We were however, not able to detect induction of measurable TNF levels in our new experiments. We have therefore considered the experiment where we saw TNF induction as an erroneous experiment, where TNF induction might have been caused by a contamination. These data were removed and replaced with our new data.

- Figure 4E: lanes 665-667 “In agreement with this, the *C. albicans* strains *efg1 Δ / Δ* *cph1 Δ / Δ* and *hgc1 Δ / Δ* [...] induced no or very little IL-1b secretion”; the panel shows similar values between these strains and *ece1 Δ / Δ*

Reply:

The reviewer is correct that the *hgc1 Δ* mutant indeed induced similar levels of IL-1 β release as the *ece1 Δ* mutant, however, the *efg1 Δ / cph1 Δ* double knock out induced significantly lower levels of IL-1 β release as the *ece1 Δ* mutant. In contrast to the *efg1 Δ /cph1 Δ* double knockout, which does not

form hyphae and does not express *ECE1*, the *hgc1Δ* mutant can to some extent still produce Candidalysin. We have clarified this in the text (**lines 298-306** of the revised manuscript).

The major message of this figure, however, is that both hyphal formation as well as *ECE1* expression induce IL-1 β release, that a combination of both causes the highest levels of IL-1 β and that supplementation of exogenous Candidalysin can restore IL-1 β release.

G LPS-primed human MDMs, 5 h

- lanes 684-685: “we demonstrate that Candidalysin is required for the full damage potential of *C. albicans*”. As reported by the authors, the difference between WT and *ece1Δ/Δ+ECE1 Δ 184-279* is not significant, so the sentence should be modified.

Reply:

The reviewer is correct that the *ece1Δ/Δ+ECE1 Δ 184-279* mutant indeed does not yield a statistical significant reduction of cell damage as observed by LDH release. However, we hope that the reviewer appreciates that with a p value of 0.0763 there is a strong trend towards reduced damage potential. Furthermore in the following graphs with the dynamics of cell death it can be observed that the dynamics of cell lysis of *ece1Δ/Δ* and *ece1Δ/Δ+ECE1 Δ 184-279* are highly similar.

Nevertheless for correctness we changed the sentence to “we demonstrate that loss of the *ECE1* gene is associated with a loss of the full damage potential of *C. albicans*” in the revised manuscript.

- lanes 786-787: “*C. albicans*-induced mBMDM damage required LPS-priming and was reduced after caspase-1 inhibition (Figure 6C)”: are the differences statistically significant?

The differences are not statistically significant; we have therefore changed the statement (**lines 370-371** in the revised manuscript).

Minor (technical)

- panel 1E: font size is too low and very difficult to read; moreover, the third column of each graph of this panel should refer to *ece1Δ/Δ +ECE1 Δ 184-279* according to the main text

We increased the font and graph size of the graph. The third column in the graph represents the data of the *ece1Δ/Δ +ECE1* revertant, not the *ece1Δ/Δ+ECE1_{Δ184-279}* knockout. We clarified this in the text.

- panel organization in some figures is questionable. For instance, Figure 1 is related to human MDMs with the exception of panel D, which refers to murine BMDMs. Murine BMDMs are addressed in Figure 2 and, in my opinion, panel D is more appropriate in this Figure.

We reorganized the panels in such a way that former figure 1 (now **Figure 2** in the revised manuscript) contains all the human data panels and former figure 2 (now **Figure 3** in the revised manuscript) contains all the murine data panels.

- **Suppl Figure 1C:** could the authors provide statistical significance between untreated and Candidalysin?

Statistical tests have been included and show a significant IL-1 β induction for priming by Zymosan and Curdlan.

- Figure 3D (human MDMs): caspase 1 p20 is difficult to assess.

We agree that in the blot it is somewhat difficult to see the caspase-1 p20 bands. Although we did not observe any bands using the *ece1Δ/Δ* mutant and stronger bands using the Candidalysin peptide. The fact that the blot was difficult to read made us use additional methods to quantify caspase-1 activity in hMDMs (see above).

References:

- 1 Casadevall, A. & Pirofski, L. A. Microbiology: Ditch the term pathogen. *Nature* **516**, 165-166, doi:10.1038/516165a (2014).
- 2 Hellmann, A. M. *et al.* Human and Murine Innate Immune Cell Populations Display Common and Distinct Response Patterns during Their In Vitro Interaction with the Pathogenic Mold *Aspergillus fumigatus*. *Front Immunol* **8**, 1716, doi:10.3389/fimmu.2017.01716 (2017).
- 3 Moyes, D. L. *et al.* Candidalysin is a fungal peptide toxin critical for mucosal infection. *Nature* **532**, 64-68, doi:10.1038/nature17625 (2016).
- 4 Gresnigt, M. S. *et al.* Neutrophil-mediated inhibition of proinflammatory cytokine responses. *J Immunol* **189**, 4806-4815, doi:10.4049/jimmunol.1103551 (2012).
- 5 Lionakis, M. S., Lim, J. K., Lee, C. C. & Murphy, P. M. Organ-specific innate immune responses in a mouse model of invasive candidiasis. *J Innate Immun* **3**, 180-199, doi:10.1159/000321157 (2011).
- 6 Medzhitov, R., Schneider, D. S. & Soares, M. P. Disease tolerance as a defense strategy. *Science* **335**, 936-941, doi:10.1126/science.1214935 (2012).
- 7 White, F. F., Yang, B. & Johnson, L. B. Prospects for understanding avirulence gene function. *Curr Opin Plant Biol* **3**, 291-298 (2000).
- 8 Verma, A. H. *et al.* Oral epithelial cells orchestrate innate type 17 responses to *Candida albicans* through the virulence factor candidalysin. *Sci Immunol* **2**, doi:10.1126/sciimmunol.aam8834 (2017).
- 9 Richardson, J. P. *et al.* Processing of *Candida albicans* Ece1p Is Critical for Candidalysin Maturation and Fungal Virulence. *MBio* **9**, doi:10.1128/mBio.02178-17 (2018).
- 10 Cullen, S. P., Kearney, C. J., Clancy, D. M. & Martin, S. J. Diverse Activators of the NLRP3 Inflammasome Promote IL-1 β Secretion by Triggering Necrosis. *Cell Rep* **11**, 1535-1548, doi:10.1016/j.celrep.2015.05.003 (2015).
- 11 Uwamahoro, N. *et al.* The pathogen *Candida albicans* hijacks pyroptosis for escape from macrophages. *MBio* **5**, e00003-00014, doi:10.1128/mBio.00003-14 (2014).

Reviewers' Comments:

Reviewer #2:

Remarks to the Author:

The revised manuscript presents convincing data that answer my questions from the first review. The conclusions in the manuscript are strongly supported by the data.

The Abstract needs to be revised in regards to "...physical forces of glucose-consuming hyphae" (line 69) – the recent glucose competition study shows that the key event that triggers macrophage death in later phases of infection is depletion of glucose by *Candida*, not so much the physical damage to macrophages by growing hyphae. I suggest rephrasing to "... in addition to damage caused by pyroptosis and glucose depletion by the growing hyphae".

Reviewer #3:

Remarks to the Author:

All my comments have been properly addressed.
I have no further comments.

Response to reviewers' comments

Reviewer #2 (Remarks to the Author):

The revised manuscript presents convincing data that answer my questions from the first review. The conclusions in the manuscript are strongly supported by the data.

The Abstract needs to be revised in regards to "...physical forces of glucose-consuming hyphae" (line 69) – the recent glucose competition study shows that the key event that triggers macrophage death in later phases of infection is depletion of glucose by *Candida*, not so much the physical damage to macrophages by growing hyphae. I suggest rephrasing to "... in addition to damage caused by pyroptosis and glucose depletion by the growing hyphae".

We agree with reviewer 2 that based on recently published data, glucose consumption by growing hyphae seem to play a major role during macrophage damage caused by *C. albicans* while the role of physical forces is less clear. We have therefore now deleted the terms "physical damage" and "physical forces" from the abstract and rephrased as following:

Clearance of invading microbes requires phagocytes of the innate immune system. However, successful pathogens have evolved sophisticated strategies to evade immune killing. The opportunistic human fungal pathogen *Candida albicans* is efficiently phagocytosed by macrophages, but causes inflammasome activation, host cytolysis, and escapes after hypha formation. Previous studies suggest that **macrophage lysis by *C. albicans* results from early inflammasome-dependent cell death (pyroptosis), late damage due to glucose depletion and membrane piercing by growing hyphae.** Here we show that Candidalysin, a cytolytic peptide toxin encoded by the hypha-associated gene *ECE1*, is both a central trigger for NLRP3 inflammasome-dependent caspase-1 activation via potassium efflux and a key driver of inflammasome-independent cytolysis of macrophages and dendritic cells upon infection with *C. albicans*. This suggests that Candidalysin-induced cell damage is a third mechanism of *C. albicans*-mediated mononuclear phagocyte cell death **in addition to damage caused by pyroptosis and by growth of glucose-consuming hyphae.**